# Mediator 1 ablation induces enamel-to-hair lineage conversion in mice through enhancer dynamics

Roman Thaler[1,2], Keigo Yoshizaki[3], Thai Nguyen[4], Satoshi Fukumoto[5,6], Pamela Den Besten[7], Daniel D. Bikle [4] & Yuko Oda [4✉]

Postnatal cell fate is postulated to be primarily determined by the local tissue micro-environment. Here, we find that Mediator 1 (*Med1*) dependent epigenetic mechanisms dictate tissue-specific lineage commitment and progression of dental epithelia. Deletion of *Med1*, a key component of the Mediator complex linking enhancer activities to gene transcription, provokes a tissue extrinsic lineage shift, causing hair generation in incisors. *Med1* deficiency gives rise to unusual hair growth via primitive cellular aggregates. Mechanistically, we find that MED1 establishes super-enhancers that control enamel lineage transcription factors in dental stem cells and their progenies. However, *Med1* deficiency reshapes the enhancer landscape and causes a switch from the dental transcriptional program towards hair and epidermis on incisors in vivo, and in dental epithelial stem cells in vitro. *Med1* loss also provokes an increase in the number and size of enhancers. Interestingly, control dental epithelia already exhibit enhancers for hair and epidermal key transcription factors; these transform into super-enhancers upon *Med1* loss suggesting that these epigenetic mechanisms cause the shift towards epidermal and hair lineages. Thus, we propose a role for *Med1* in safeguarding lineage specific enhancers, highlight the central role of enhancer accessibility in lineage reprogramming and provide insights into ectodermal regeneration.

[1] Department of Orthopedic Surgery, Mayo Clinic, Rochester, MN, USA. [2] Center for Regenerative Medicine, Mayo Clinic, Rochester, MN, USA. [3] Section of Orthodontics and Dentofacial Orthopedics, Division of Oral Health, Growth and Development, Kyushu University Faculty of Dental Science, Fukuoka, Japan. [4] Departments of Medicine and Endocrinology, University of California San Francisco and San Francisco Veterans Affairs Health Center, San Francisco, CA, USA. [5] Section of Pediatric Dentistry, Division of Oral Health, Growth and Development, Kyushu University Faculty of Dental Science, Fukuoka, Japan. [6] Division of Pediatric Dentistry, Department of Oral Health and Development Sciences, Tohoku University Graduate School of Dentistry, Sendai, Japan. [7] Department of Dentistry, University of California San Francisco, San Francisco, CA, USA. ✉email: yuko.oda@ucsf.edu

Postnatal cell fates are controlled by selective transcriptional programs, that are coordinated by an underlying epigenetic machinery and cellular microenvironments[1–4]. Ectoderm evolves into multiple lineages including enamel forming dental lineages, or to hair and epidermis generating skin epithelia[5–7]. Adult stem cells are endowed with tissue-specific potential maintaining however the ability to differentiate into different cell types[8–10]. To date, the mechanisms controlling specific cell lineage commitment are yet not fully understood, and accordingly, the factors safeguarding dental or epidermal cell lineages are still largely elusive. Understanding these mechanisms is also crucial for our efforts in combating diseases like alopecia or tooth loss as it might create entry points for the development of bioengineering strategies for the regeneration of tissues like tooth enamel and skin hair, which have been challenging[7].

The mouse incisor provides an excellent model system to study postnatal lineage commitment and progression. Dental epithelial stem cells (DE-SCs) residing in the cervical loop (CL) at the proximal end of incisors support the continuous growth of the incisors throughout the lifespan of a mouse. In contrast, dental epithelia on molars are not regenerated once molars are developed. DE-SCs regenerate the enamel organ on incisor by giving rise to all the dental epithelia including enamel matrix producing ameloblasts, Notch1 regulated stratum intermedium (SI), and others[11–16].

Skin epithelia are derived from the same ectoderm and are divided into the interfollicular epidermis and the hair follicles during embryonic stages of skin formation[6]. Postnatally, hair follicle stem cells residing in the bulge region of hair follicles regenerate hair in the skin[17,18]. Hair follicles are essential for hair growth and regeneration in mammals and are characterized by a complex multilayered bulb structure containing the outer root sheath (ORS), the companion layer (Cl), the inner root sheath (IRS), and the medulla/cortex; all these structures are characterized by the selective expression of different hair keratins[19,20], such as Keratin (KRT) 75 which labels Cl, KRT71 found in IRS, and KRT31 which is expressed in hair cortex, as well as the epithelial marker KRT14 which marks ORS. Following the initial developmental cycle, hair is maintained by hair follicle cycling in mice. This is characterized by an anagen phase during which the hair follicle expands, followed by a transitional catagen phase which leads to the involution of the hair follicle, and telogen, the resting phase. The cycle then begins a new in response to signals from the dermal papilla, a specialized mesenchymal group of cells attached to the proximal end of the hair bulb[21,22]. The shift between these phases involves substantial changes in gene expression patterns, which can be activated by injury. For example, hair depilation induces new hair growth by stimulating the anagen phase with induction of genes such as those encoding hair keratins[23].

Several transcription factors and pathways have been shown to induce dental as well as hair and epidermal cell differentiation. During embryonic development, dental and skin epithelia share similar transcriptional networks as they are derived from the same ectoderm. Postnatally however, each lineage is controlled by specific transcription factors. For example, paired like homeodomain 2 (*Pitx2*)[24], Nk homeobox 3 (*Nkx2-3*)[25], and ISL LIM homeobox 1 (*Isl1*)[26] are implicated in enamel epithelial homeostasis[7] while transcription factors and signaling genes including HR lysine demethylase and nuclear receptor corepressor (*Hr also called hairless*) and Ras and Rab interactor 2 (*Rin2*) are important in controlling hair differentiation as mutations in these genes cause alopecia (hair loss)[27,28]. On the other hand, AP1 factors (*Fos/Jun*) and the tumor proteins of the p53 family including *Tp63* are crucial to direct postnatal epidermal differentiation in the skin[20]. Nevertheless, the check point factors specifying dental vs epidermal cell fate are still largely elusive.

The Mediator complex controls cell lineage. MED1 is a functional subunit of the Mediator complex, that facilitates gene transcription by linking enhancers to the RNA polymerase complex at transcriptional start sites (TSS) in gene promoters. MED1 has been shown to localize at typical enhancers (TE) and super-enhancers (SE) which represent distal regulatory elements recognized as key epigenetic hubs to determine cell identity and properties[4,29]. As such, the Mediator complex facilitates the expression of cell fate driving transcription factors and genes. In fact, Med1 assures pluripotency of embryonic stem (ES) cells by enforcing the expression of *Klf4, Oct4, Sox2,* and *Nanog*, four key reprogramming factors. In addition, reduced expression of Mediator subunits like *Med1* triggers differentiation of ES cells[4,29]. Deletion of *Med1* also affects somatic cell fates such as shown in epidermal cells in skin[30–34]. *Med1* ablation hinders invariant natural killer T cell development[35] and prevents luminal cell maturation in the mammary epithelia in mice[36,37]. These data indicate a role for *Med1* in controlling lineage commitment in somatic cells as well. However, the exact role of *Med1* is not fully understood as previous biochemical studies indicated that removal of Med1 disturbs selective transcription[38], but is dispensable for Mediator complex formation[38].

In the present study, we investigate the role of *Med1* in governing ectoderm lineage development. Using a conditional knockout (cKO) mouse for *Med1*, we show that ablation of *Med1* in ectoderm derived epithelial cells causes hair generation on incisors while they lose their ability to produce enamel resulting in severe enamel dysplasia[33,39,40]. We describe a unique cellular arrangement by which *Med1* ablation causes ectopic hair growth on incisors, and using an integrative multi-omics approach we demonstrated that this intriguing phenomenon is due to a reshape of the enhancer landscape in which ectoderm conserved enhancers are amplified to induce epidermal and hair driving transcription factors.

## Results

**Lack of *Med1* in dental epithelia causes ectopic hair on incisors**. During postnatal murine tooth maintenance, CL residing dental epithelial stem cells (DE-SCs) differentiate into Notch1 regulated SI, ameloblasts, and other epithelia to mineralize enamel on the incisor surface (Fig. 1a, top). Intriguingly, deletion of *Med1* from *Krt14* expressing DE-SCs and their progenies provoked a major phenotypical shift in the dental compartment causing enamel formation to be replaced by unusual, ectopic hair growth on mouse incisors but not on molars as we have previously reported[33]. This phenomenon has also been observed in two further unrelated mouse models[41,42] and has been described in a few clinical cases[43]. As hair growth on the skin depends on multiple cell populations arranged to form hair follicles (Fig. 1a, bottom), the question arises on how hair development is achieved in a dental environment. Therefore, we evaluated the cellular processes leading to hair development in *Med1* cKO incisors by comparing it with physiological hair in skin. Although hair grown on incisors of our mouse model showed a comparable hair architecture (cuticle), composition (guard hair and zig-zag hair), and morphology as naturally grown skin hair (Fig. 1b), hair formation significantly differed on incisors in which hair supporting tissues and root structures lacked hair follicles (Fig. 1b). In fact, while skin grown hair is regenerated from large anagen follicles in response to signals from the dermal papilla and maintained by small telogen follicles (Fig. 1c, bottom orange arrows), we found that dental hair is surrounded by atypical cell clusters without typical hair follicle-like structures in *Med1* cKO incisors (Fig. 1c top, yellow triangles, and Supplementary Fig. 1e, blue arrows with dotted lined region). These abnormal cell clusters

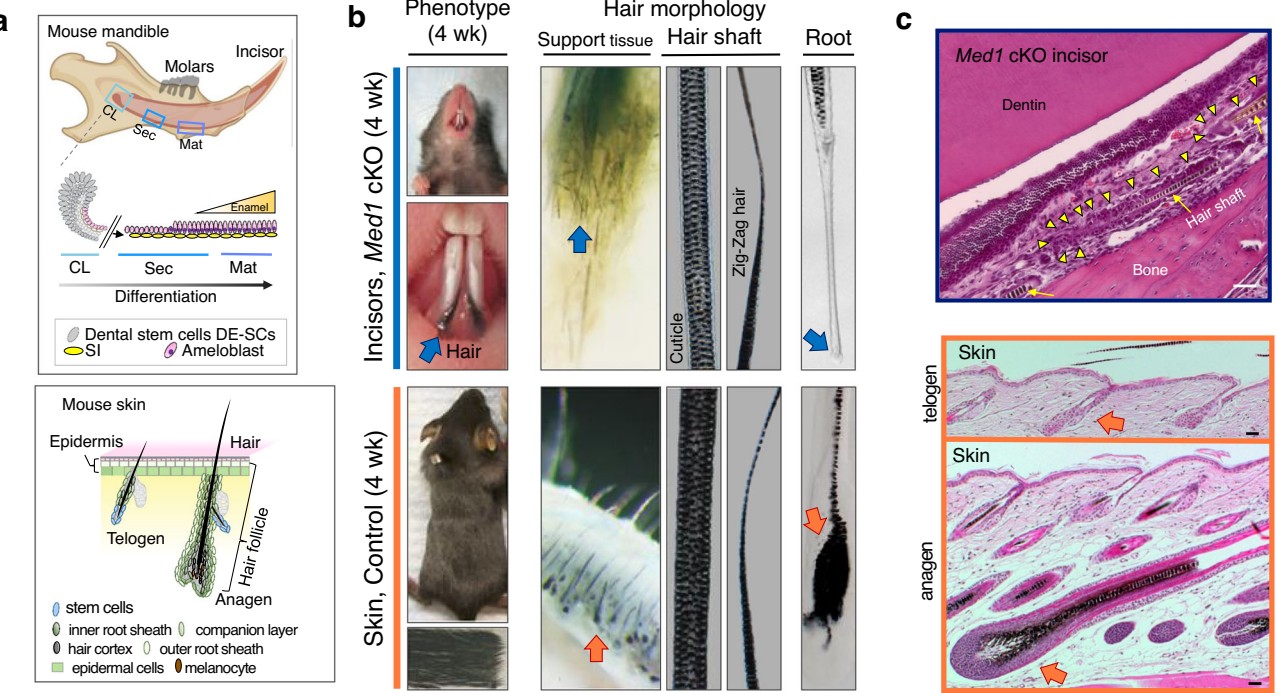

**Fig. 1 Loss of *Med1* in dental stem cells causes ectopic hair growth on incisors. a** Top, diagram representing normal dental epithelial differentiation and enamel formation in mouse mandible. CL cervical loop, Sec secretory stage, Mat maturation stage, DE-SCs dental epithelial stem cells, SI stratum intermedium. Bottom, hair regeneration by hair follicles under hair cycling regulation of resting telogen (left) and growing anagen (right) in skin. **b** Hair growth on *Med1* cKO incisors at 4 weeks of age (top) is compared to normal hair in skin (bottom). Arrows show the root of hairs. **c** Top, HE sections of tissues emphasizing the atypical cell cluster (yellow triangles) surrounding hair shafts (yellow arrow) found between dentin and bone in *Med1* cKO incisor (3 month). Bottom, HE sections for hair follicles at anagen (bottom) and telogen (top) phases supporting hair growth and regeneration in the skin (orange arrows). Bars = 50 µm. For **b**, **c**, representative images are shown. The diagram depicting the mouse mandible (brown colored) was created with BioRender.com, as were the ones shown in Figs. 2a, 3a, c, and 4a and Supplementary Figs. 1a and 4a.

originated from a disorganized SI layer (Supplementary Fig. 1a, b, red triangles) that progressed into expanded papillary layers (Supplementary Fig. 1c, yellow triangles) to form the hair bases in dental mesenchymal tissues (Supplementary Fig. 1e, blue triangles). NOTCH1 positive SI/SR derived papillary cells formed unusual cell aggregates and differentiated into the hair lineage as demonstrated by the expression of the hair follicle marker KRT71 at 4 weeks of age (Supplementary Fig. 1d). At adult stages (3 months of age and later), these aggregates formed aberrant pocket-like structures (Fig. 2a). Although the NOTCH1 expressing SI/SR derived cell clusters gave rise to cells expressing the hair marker KRT71 and the epidermal marker Loricrin (LOR) at 3 months of age (Fig. 2a, b), the spatial organization of these cells was scattered and randomly distributed (Fig. 2b two left panels, arrows). This clearly differs from the well-defined cellular framework found along hair follicles and on the surface of the interfollicular epidermis in the skin (Fig. 2b, right panels, orange arrows). Furthermore, hair depilation of skin initiated the hair cycle by enlarging hair follicles (anagen) (Fig. 2c left 3 panels, orange arrows) and by inducing the expression of hair keratin *Krt31* (Supplementary Fig. 2a, skin). In contrast, dental hair depilation did not activate any follicle-like structures in the hair supporting tissues on *Med1* null incisors (Fig. 2c two right panels, blue arrows), in either distal or proximal incisor regions (Supplementary Fig. 2b) nor was the expression of the hair marker *Krt31* induced (Supplementary Fig. 2a tooth). Nevertheless, hair grew back at its original length 12 days after depilation in *Med1* cKO incisors (Fig. 2d blue arrow), as occurred in normal skin (Supplementary Fig. 2c). These results demonstrate that hair can grow and is regenerated through SI-derived dental epithelia (Fig. 2e).

To better understand dental hair growth and regeneration, we compared hair follicles of the skin, including distinct hair follicle layers, mesenchymal cells such as dermal papilla, and the presence of melanocytes to pigment the hair, with the cell clusters found around dental hair in *Med1* cKO mice.

In the skin, hair follicles have distinct cellular layers. Using 4 different antibodies, these layers were well distinguishable in cross sections of normal skin hair follicles (4 weeks, anagen). They form ring structures in which the ORS layer (KRT14, red) was outmost, followed by a KRT75 positive Cl layer and an internal KRT71 expressing IRS layer (Supplementary Fig. 3b left 3 panels) as shown in the diagram in Supplementary Fig. 3a). The innermost layer was positive for KRT31 (Supplementary Fig. 3b). The arrangement and spatial distancing from KRT14 positive cells were also observable in sagittal sections (Supplementary Fig. 3b far right panels), as shown in the diagram (Supplementary Fig. 3c). Next, we assessed if these organized structures can also be found around the bases of dental hair in *Med1* cKO mice. Here, cells expressing KRT75, KRT71, or KRT31 were found as well but they were scatted and did not form follicle structures (Supplementary Fig. 4a, b), thus clearly differing from the skin. Also, no spatial distancing from KRT14 positive cells was observable in sagittal sections (Supplementary Fig. 4b, yellow triangles in upper panels) nor were ring-like structures found at cross sections (Supplementary Fig. 4a, b lower panels).

These results were further confirmed when cryosections of hair generating tissue dissected from *Med1* cKO mandibles (Supplementary Fig. 5a) were analyzed. Again, three hair keratins (KRT75, KRT71, KRT31) were distributed randomly near hair shafts, but did not organize into hair follicle structures (Supplementary Fig. 5b).

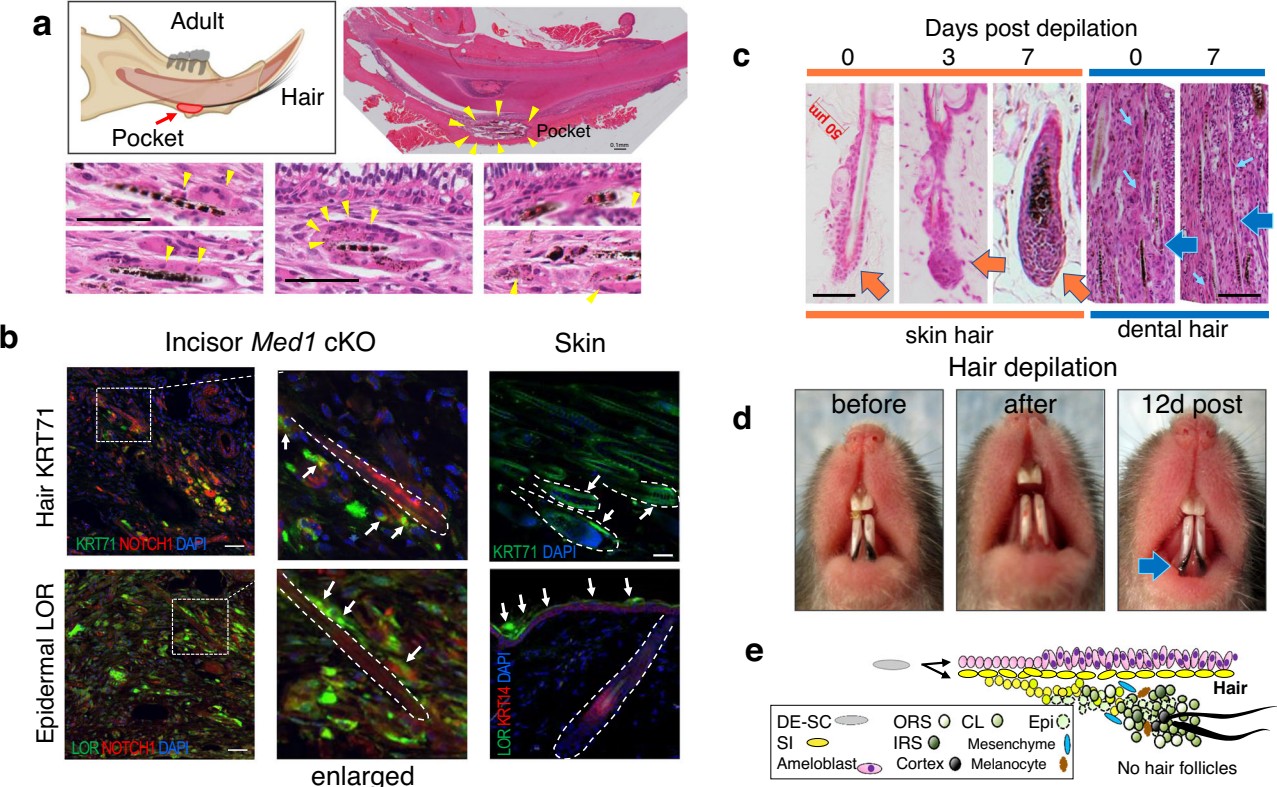

**Fig. 2 *Med1* null mice develop and regenerate hair on incisors. a** Left, diagram depicting hair generation via pocket-like structures (red) in *Med1* cKO incisor in adult mice. Right and lower picture set, HE staining to evidence hair in pocket formation and eosin-positive aberrant cell clusters surrounding hair shafts (yellow triangles) in dental tissues. **b** Top, immuno-staining for hair marker KRT71 (green) and dental SI marker NOTCH1 (red) in the hair-generating tissues in *Med1* cKO incisors or for KRT71 only (green) in normal skin. Bottom, epidermal marker LOR (green) and NOTCH1 (red) in dental tissues compared to LOR (green) and KRT14 (red) localization in the skin. The location of hair shafts is marked by dotted lines. **c** HE staining on sections of skin (orange) and dental (blue) tissues before (day 0) and after hair depilation (3 and 7 days). Large orange arrows show hair follicles in the skin. Large blue arrows show hair roots in dental tissues. Small pale blue arrows show eosin positive cell clusters. Bars = 50 μm. **d** Full hair regeneration 12 days after hair depilation from *Med1* cKO mice (blue arrow). **e** Schematic representation of cellular processes and anatomical location of dental SI/SR derived dental epithelia (yellow) that are gradually transformed into epidermal (dotted circle) and various hair gene expressing cells (solid circles). Multiple hair keratins are expressed, equivalent to skin hair follicle layers including companion layer (Cl), inner root sheath (IRS), outer root sheath (ORS), and hair cortex, but in disorganized manner in aberrant cell aggregates that lack hair follicle structures in *Med1* cKO incisors. Presence of mesenchymal cells and melanocytes are also shown in blue and brown, respectively. For **a-d**, all of the *Med1* cKO mice have the same phenotypes, and representative images are shown, and reproducibility was confirmed at least in two different litters of *Med1* cKO and control mice (*n* = 3).

Regarding the presence of dermal papilla, such structures were also missing in the roots of incisor-grown hair lacking hair bulbs (HE stains in Fig. 1c, and Supplementary Fig. 1a–e). Nevertheless, we found the mesenchymal protein VIMENTIN (red) in KRT71 expressing hair generating cell clusters in *Med1* cKO mice (6 months)(Supplementary Fig. 5c yellow triangles). Although VIMENTIN does not represent a classic marker for dermal papilla, it was localized in this structure in the skin[44]. Again, VIMENTIN expressing cells were scattered and did not form well organized structures, suggesting lack of papilla-like structures in *Med1* cKO mice.

Next, we assessed melanin accumulation in the pocket tissues of *Med1* cKO incisors which generated well-pigmented black hair (C57BL6 background) (Supplementary Fig. 6a, yellow arrows). Brown-colored melanin-producing cells were randomly distributed (Supplementary Fig. 6a, enlarged images) again differing from the skin in which the melanin is discretely accumulated inside of the hair bulbs (Supplementary Fig. 6b, 4 week anagen). In addition, mRNA expressions of melanocyte markers were elevated in incisors of *Med1* cKO mice (Supplementary Fig. 6c, RNA-seq) indicating the presence of melanocyte-like cells. It is plausible that this type of cell might be generated through an

aberrant lineage conversion of neural crest cells, multipotent stem cells that regenerate mandibular cells including dental mesenchymal cells[45].

Altogether, these results demonstrate that *Med1* cKO incisors generate all the components needed for hair growth and regeneration as found in the skin, but they fail to organize them in well-defined hair follicle structures. Expression of hair and epidermal markers in the hair generating dental cell aggregates further indicates that the presence of certain transcriptional patterns allow for the generation of hair via a primitive cellular environment.

**Med1 deficiency is sufficient to switch the dental transcriptional program towards hair and epidermis in incisors in vivo.** Cell identity and cell lineage progression tightly associate with a timed expression of specific gene-sets. As loss of *Med1* causes hair growth in the dental compartment, using total gene expression datasets we investigated the expression patterns of epidermal and hair-related gene sets at distinct anatomical locations of incisors in *Med1* cKO mice (Fig. 3a, diagram). Our analyses revealed that, compared to control mice, in *Med1* cKO mice some epidermal

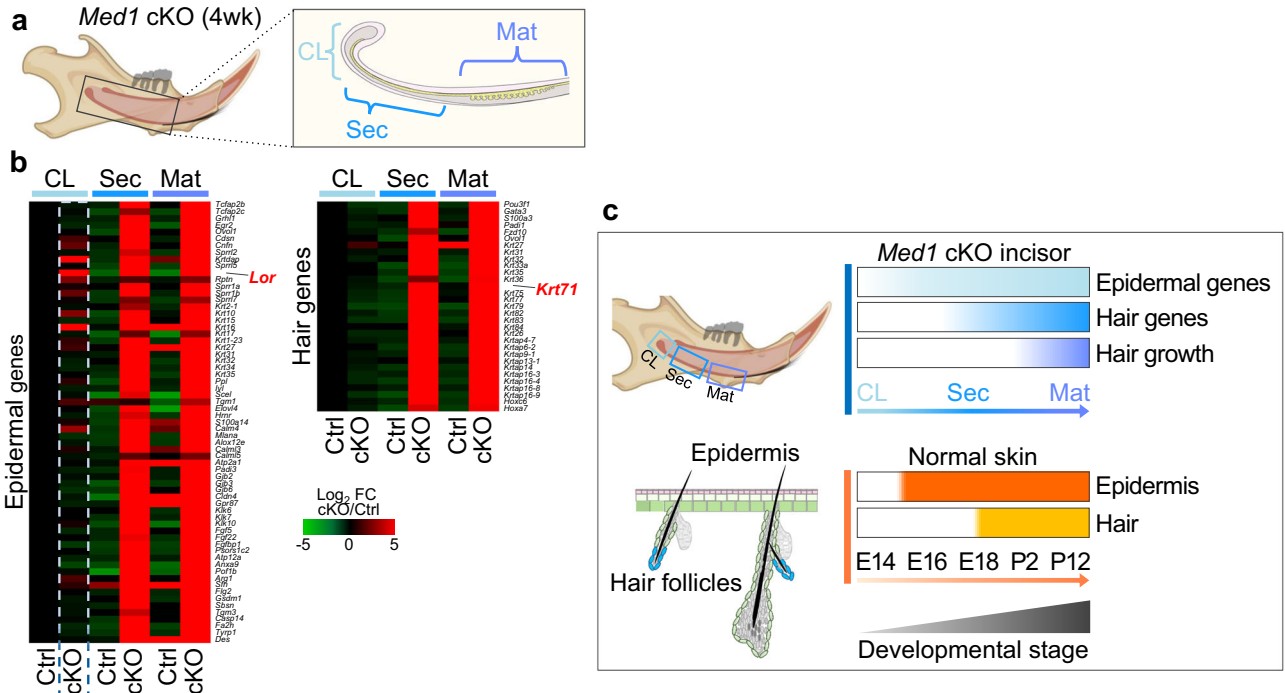

**Fig. 3 Loss of *Med1* activates epidermal and hair gene expression in developing incisors. a** Diagram depicting the anatomical locations tested in Ctrl and *Med1* cKO incisors. CL, cervical loop; Sec, secretory stage; Mat, maturation stage. **b** Heatmap showing differential gene expressions for epidermal genes (left) and hair genes (right) in *Med1* cKO incisors versus Ctrls at 4 weeks of age. For each gene, fold changes are compared to CL Ctrl; *n* = 3 for all samples and average values for each group are shown. **c** Top, diagram depicts sequential expression of epidermal genes (light blue) and hair genes (blue) as well as hair growth (purple) in *Med1* cKO mice regarding the different anatomical locations on incisors (CL, Sec, and Mat) of *Med1* cKO incisors. Bottom, pattern comparison to epidermal (orange, E15) and hair gene inductions (yellow, E18) during embryonic development of the skin.

genes are already induced in the DE-SCs containing CL region at 4 weeks of age (Fig. 3b left panel, blue dotted box, Fig. 3c pale blue bar). Subsequently, as DE-SCs differentiate towards the secretory (Sec) and maturation (Mat) stages, expression of many epidermal genes strongly increase in *Med1* deficient tissue (Fig. 3b). Intriguingly, hair-related genes were not yet detected in the CL locus but become clearly upregulated downstream in the Sec and Mat regions (Fig. 3b right panel, Fig. 3c blue bar). Different hair keratin genes representing distinct layers of hair follicles were upregulated, including KRT75 for supportive companion layer, KRT31 for hair cortex, and KRT71 for IRS (Fig. 3b right panel). Immunostaining confirmed the sequential induction of epidermal and hair markers as the epidermal markers LOR and KRT1 were detected in the secretory stage as we have previously described[39] but KRT71 was expressed only at the maturation stage (Supplementary Fig. 1d). These data illustrate how in cKO mice the expression of selected gene-sets during dental hair development parallels embryonic and postnatal skin development in wild-type mice, where epidermal development occurs first (E15) (Fig. 3c orange bar), followed by hair follicle formation (E18) (Fig. 3c yellow bar). These results demonstrate that *Med1* deficiency is sufficient to implement epidermal and hair-related transcriptional programs priming dental stem cells towards skin epithelial differentiation in vivo.

***Med1* deletion redirects dental epithelial stem cells towards an epidermal fate in vitro.** Although deletion of *Med1* causes a lineage shift in vivo, the question arises if aberrant dental hair growth is due to lineage predisposition of dental epithelial stem cells. To investigate this phenomenon, dental stem cells from CL tissues were cultured. CL tissues from 8–10 weeks old control and *Med1* cKO mice were micro-dissected and digested cells were plated until self-renewing stem cell colonies formed (Fig. 4a, b). An enhanced proliferation rate was observed for the *Med1* deficient

cells as shown by the increased number of BrdU positive cells as well as by the generation of larger colonies (Fig. 4b, cKO). Confirming and expanding these results, IPA (ingenuity pathway analysis) of whole transcriptomic data from these cultures not only revealed an upregulation of cell growth and proliferation related pathways in *Med1* null cells, but also showed that these cells spontaneously differentiate into the epidermal lineage in monolayer cultures (Fig. 4c, d), even without factors or mesenchymal feeder cells to induce epidermal differentiation. In fact, like Sec tissue from cKO mice (Fig. 4d, middle column), an extensive set of epidermal-related genes are clearly upregulated in cultured *Med1* null DE-SCs as well (Fig. 4d right column). Of note, in contrast to our in vivo data shown above, genes related to hair differentiation are only partially induced in vitro (Fig. 4d), suggesting that full hair differentiation requires additional external processes. Furthermore, IPA analysis also identified members of the p53 family (Tp63/53/73) as top upstream inducers of epidermal differentiation (Fig. 4e) that are linked to the key epidermal AP-1 transcription factors *Fos* and *Jun* (Fig. 4f), resembling their roles in the skin. Collectively, these results show that Med1 deletion reprograms DE-SCs towards epidermal fates through a cell intrinsic mechanism. Together with our in vivo observation in *Med1* cKO mice, these results lead us to investigate the underlying epigenetic and transcriptional mechanism, by which Med1 assures enamel lineage and Med1 deficiency causes a cell lineage shift.

**Med1 regulates enamel lineage driving transcription factors through super-enhancers.** Lineage commitment and reprogramming are controlled by cell fate driving transcription factors. Mediator including MED1 subunit orchestrates these key factors by associating with super-enhancers[29]. Therefore, we first assessed the genome wide distribution of MED1 during dental epithelial development in control mice (4 week), then identified MED1-

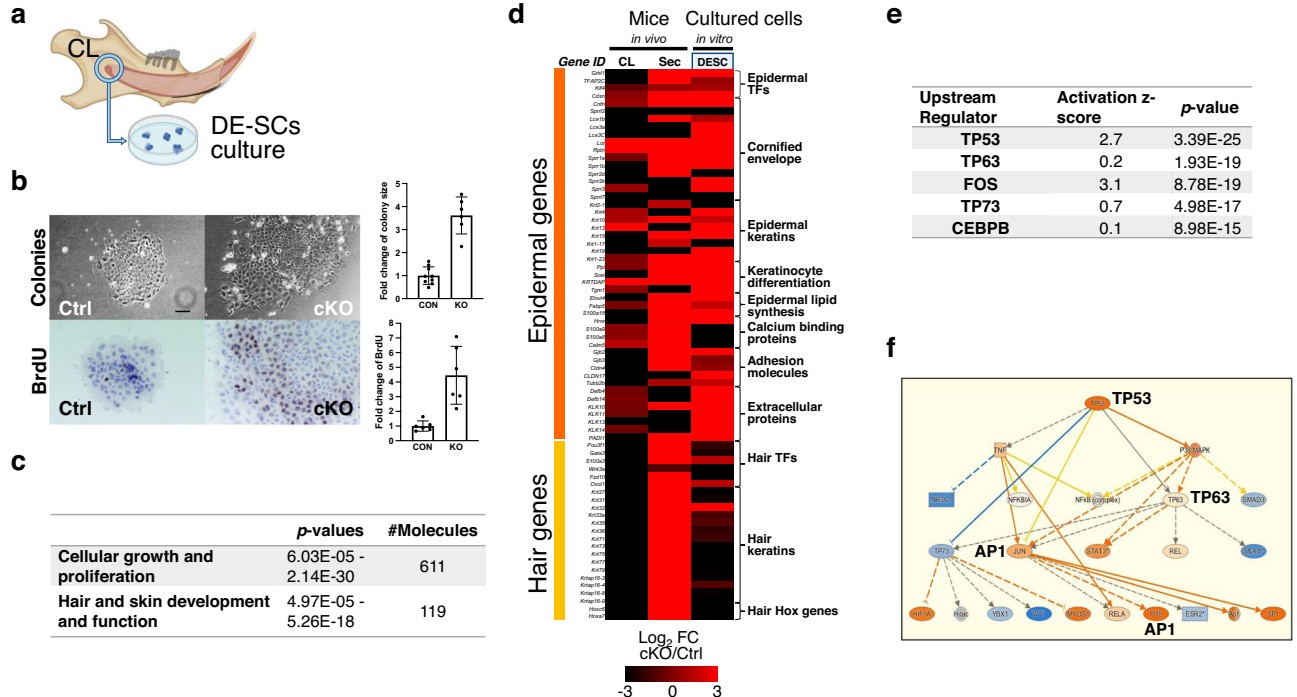

**Fig. 4 *Med1* deficiency directs DE-SCs towards epidermal fate in vitro. a** Generation of DE-SC culture from CL tissues. **b** Left, representative bright field images of DE-SC colonies and BrdU staining in Ctrl and *Med1* cKO. Bar = 10 mm. Right, quantification of colony size and BrdU positive cells/colony are shown as fold changes (cKO/Ctrl) with standard deviations (error bars); in both comparisons (n = 6–9, t-test p < 0.01). **c** Biological processes associated with loss of *Med1* in DE-SCs as identified by Ingenuity Pathway Analysis (IPA) on microarray gene expression data. **d** Heatmap showing differential gene expressions of epidermal and hair-related genes in *Med1* cKO vs Ctrl incisor tissues (CL and Sec in vivo) and cultured *Med1* cKO vs Ctrl DE-SCs (third lane); n = 3 for all samples and average values for each comparison are shown. **e** Upstream regulators responsible for biological process induced by loss of *Med1* in DE-SCs as identified by IPA on microarray data as used in **c**. **f** Mechanistic network representation for TP53/63 pathways in induced in *Med1* lacking DE-SC culture regulating epidermal fate driving AP-1 factors; upregulated genes are in orange and direct relationships are shown by solid lines. Reproducibility was confirmed by two independent cultures, and representative data are shown.

regulated transcription factors through super-enhancer analyses. For this purpose, we performed MED1 ChIP-seq of the stem cell containing head region of the CL (CLH) as well as of the CL tail area (CLT) which includes the stem cell progenies (Fig. 5a, left diagram). MED1 peaks were found in the distal intergenic regions, whereby an accumulation of peaks was observable in CLT vs CLH tissues (Supplementary Fig. 7a). MED1 peaks associated with several hundred super-enhancers in the CLH and CLT tissues (Fig. 5b and Supplementary Fig. 7b). These include super-enhancers at genomic loci near lineage driving transcription factors. In the CLH tissues, these encompass enamel lineage transcription factors like *Pitx2, Isl1, and Nkx2-3* (Fig. 5b left panel and Fig. 5c left 3 panels), which may prime adult stem cells to an enamel fate, consistent with their essential roles in tooth morphogenesis during embryonic development[25,46]. In contrast, in CLT samples MED1 super-enhancers were found around key transcription factors like Satb1 homeobox 1 (*Satb1*) and Runx family transcription factor 1 (*Runx1*) and 2 (*Runx2*) which control later cellular processes like differentiation and enamel mineralization[46–48] (Fig. 5b right panel and Fig. 5c right 3 panels). Of note, the mRNA expression levels of these transcription factors were strongly reduced in CLT tissues in *Med1* cKO mice (Fig. 5d and Supplementary Fig 7d), further implicating their role in enamel formation as *Med1* null mice show severe enamel dysplasia[40]. In addition, MED1 levels were also enriched around the promoters of these transcription factors but not for ameloblast markers (Supplementary Fig. 7e), indicating that MED1 directly activates enamel fate related transcription factors rather than supporting late differentiation. These data suggest that *Med1* programs dental stem cells and their progenies to commit

and progress towards the enamel lineage by controlling the expression of lineage driving transcription factors (Fig. 5e).

Intriguingly, in *Med1* cKO CLT tissues we also found enhancers around a different set of transcription factors (Supplementary Fig. 7c, pink labeled). These are involved in epidermal differentiation of the skin and included the genes of early growth response (*Egr3*), CCAAT enhancer binding protein b (*Cebpb*), Fos like 2 (*Fosl2*) (AP-1 transcription factor subunit), Kruppel like factor 3 (*Klf3*), Forkhead box C1 (*Foxc1*), *Foxo3*, tumor protein *Tp63*, and hair fate driving factors like *Hr* and *Rin2*. Their mRNA expressions were substantially upregulated in CLT tissues of *Med1* cKO mice (Fig. 5f and Supplementary Fig. 7d), implicating their roles in inducing epidermal and hair fates in *Med1* cKO incisors.

***Med1* deficiency reshapes the enhancer landscape in DE-SCs and their progenies.** As loss of *Med1* inhibits the expression of Med1-associated enamel lineage driving transcription factors while inducing epidermal transcription factors in CL tissues, we next probed if this is linked to epigenetic changes in promoter and enhancer patterns upon loss of *Med1*. To test this hypothesis, we performed ChIP-Seq analysis against an alternative enhancer marker, histone 3 lysine 27 acetylation (H3K27ac), in CLH and CLT tissues of 4-week-old *Med1* cKO mice and their littermate controls (Fig. 6a) as MED1 is not present in the *Med1* cKO tissues. It is known that histone acetylation (H3K27ac) generally co-localizes with Mediator components such as MED1 at distal regulatory elements like typical and super-enhancers and acts as a major inducer of gene expression by increasing chromatin

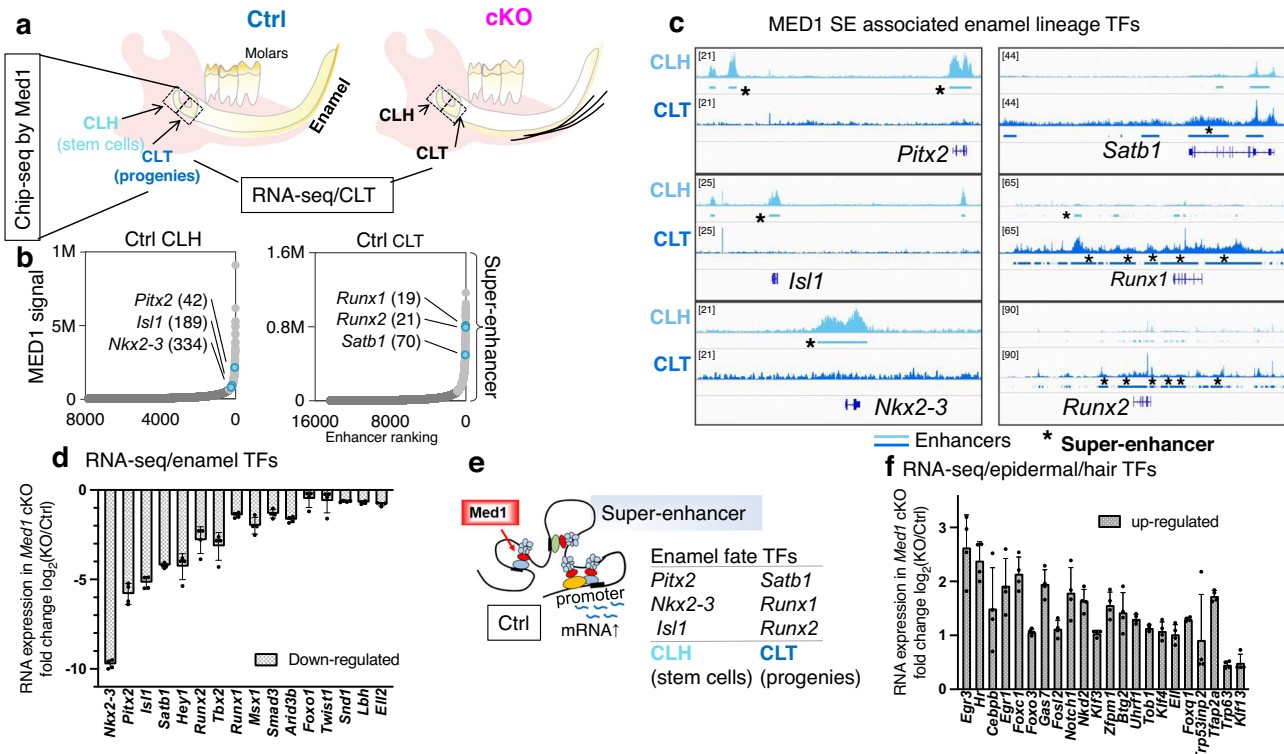

**Fig. 5 *Med1* directly controls enamel lineage transcription factors by associating with their enhancers and promoters. a** Schematic representation of tissues isolated (CLH; stem cells, CLT; stem cell progenies) from normal mouse mandibles for MED1 ChIP-seq and RNA-seq. **b** MED1 bound enhancer clustering in CLH and CLT tissues. Light gray and blue dots represent super-enhancers, dark gray dots are typical enhancers; blue dots outline super-enhancer associated with enamel transcription factors as designated by gene name and enhancer ranking numbers. **c** Genomic MED1 binding profiles in CLH (light blue) and CLT (blue) tissues on mouse genome (mm10) for relevant transcription factors. Super-enhancers (SE) are marked by bars with asterisks*. Average profiles from two independent ChIP-seq experiments are shown, in which cervical loop tissues from 2–4 mice (Med1 cKO and littermate control) are pooled for one ChIP experiment (total 4–8 cervical loop tissues). **d** Transcription factors (TFs) identified through MED1 super-enhancer that were down-regulated in *Med1 cKO* (CLT) as measured by RNA-seq. **e** Schematic of *Med1* inducing enamel fate transcription factors (TFs); distal MED1 (red) containing super-enhancers associate to gene promoters to induce mRNA expression (blue waved lines). **f** Enhancer-associated epidermal or hair fate transcription factors (TFs) that were upregulated in cKO (CLT) as measured by RNA-seq. **d**, **f** Data are shown as fold changes (log$_2$ FC) with standard deviations (n = 4, error bars) with statistical significance (t-test, p < 0.05) in combinatory analysis of cKO compared to Ctrl within two different litters of *Med1* cKO and control mice (6 CL tissues each group). The diagram of the mandibles (pink colored) is derived from our previous publication[39] but modified here, and same for ones in Fig. 6a and Supplementary Figs. 2b, 8d, and 9a, and CL tissues in Fig. 7a.

accessibility[29,49]. Compared to controls, we found major genomic changes in the H3K27ac binding patterns of *Med1* cKO CLT tissues. These were associated with biological functions such as enamel mineralization and tooth morphogenesis (Fig. 6b blue bars), as well as epidermal and hair development (yellow bars), consistent with the observations made in the *Med1* cKO phenotypes. These results suggest that phenotypic and transcriptional changes are due to epigenetic changes. In addition, *Med1* deletion substantially increased the number of super-enhancers (Fig. 6c) as well as typical enhancers (Supplementary Fig. 8a) in both, CLH and CLT tissues. Some of these new enhancers were found around loci coding for epidermal and hair lineage genes (Fig. 6c) that become upregulated upon loss of *Med1* in CLT tissue. For example, a new super-enhancer was formed at the locus coding for the epidermal transcription factor *Fosl2* (Ap-1 factor) (Fig. 6d blue bar in boxed region, Supplementary Fig. 8b), directly linking its increased mRNA expression to the loss of *Med1* (see Fig. 5f and Supplementary Fig. 7d). Corroborating these results and in line with our transcriptional network studies in *Med1* null cell cultures (see Fig. 4f), we also found that the binding motifs for the epidermal inducible transcription factors, Tp53/63, Egr1, and AP-1 are significantly enriched in super-enhancers in cKO CLT tissues compared to controls (Fig. 6e). Collectively, these data demonstrate that loss of *Med1* alters the epigenetic landscapes in

DE-SCs and their progenies. These results prompted us to investigate the epigenetic mechanism by which Med1 deficiency induces epidermal and hair lineages more in detail.

**Med1 deficiency induces epidermal and hair driving transcription factors via amplification of ectoderm conserved enhancers.** During embryonic development, dental and skin epithelia are derived from the same ectoderm by sharing transcriptional networks. Through analyses of the H3K27ac enhancer profiles of epidermal keratinocytes and hair follicle cells from published data sets[50], we further found that even though dental epithelia usually do not commit to epidermal fates, they widely share enhancers (TE/SE) with both epidermis and hair follicle derived cells for most of the transcription factors induced upon *Med1* loss (Fig. 7a). Importantly, *Med1* deficiency elevated and upgraded these shared enhancers into transcriptionally active super-enhancers as outlined for the *Hr* (hairless) locus (Fig. 7b, pink bar and Supplementary Fig. 8c). Loss of *Med1* also expanded enhancer size for epidermal factors *Foxo3* and *Cebpb* (Supplementary Fig. 8d). In addition, enhancers near many hair-lineage related genes were elevated to super-enhancers upon the loss of *Med1* in CLT (Fig. 7c and Supplementary Fig. 8e) which coincided with an induction at their mRNA levels that were measured by RNA-seq (Fig. 7d, Supplementary Fig. 9a, b).

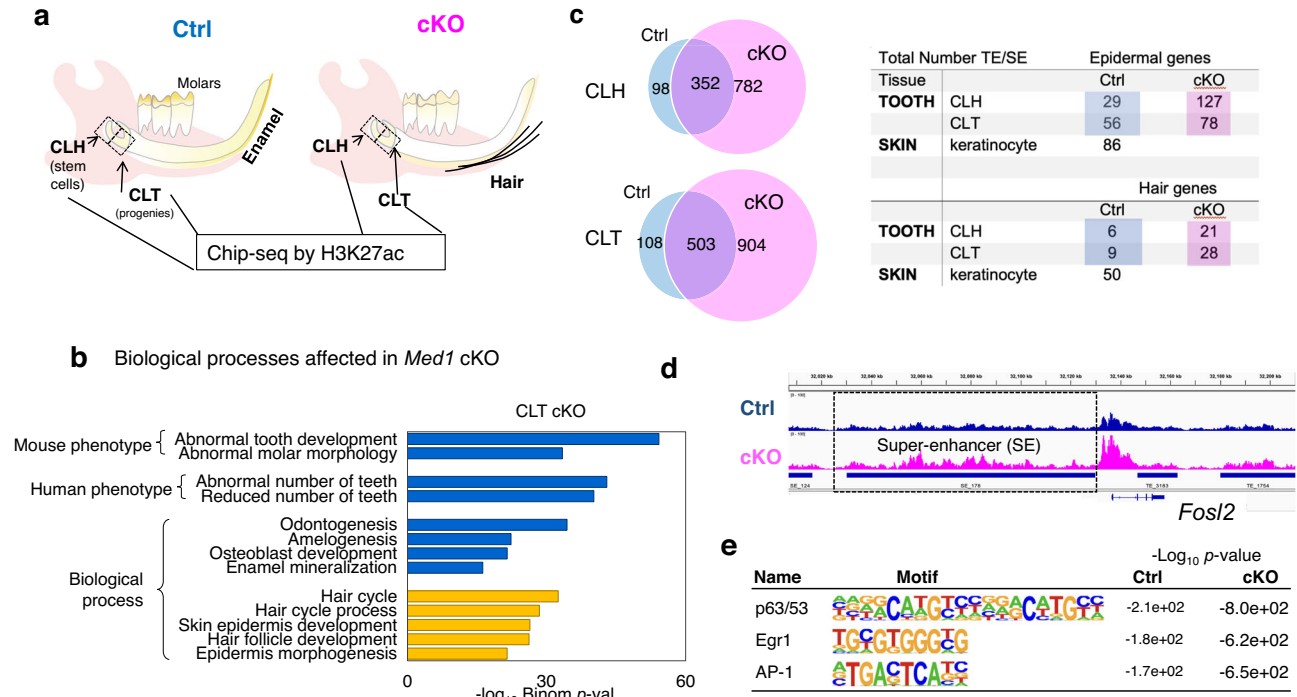

**Fig. 6 Loss of *Med1* expands the super-enhancer landscape in CLH and CLT tissues. a** Schematic representation for the isolation of CLH (stem cells) and CLT (stem cell progenies) from *Med1* cKO and littermate Ctrl mandibles for ChIP-seq against H3K27ac to compare actively transcribed genomic regions. **b** GREAT based GO analysis for genes associated with differential H3K27ac peaks to elucidate biological processes affected by *Med1* loss in CLT tissues. **c** Number of super-enhancers in Ctrl (blue) and cKO (pink) tissues. We compare typical and super-enhancers associated with epidermal and hair-related genes before and after loss of *Med1* in CLH and CLT tissues. As a comparison, data from skin-derived keratinocytes are included. **d** ChIP-seq profiles on mouse genome (mm10) for H3K27ac occupancy for the *Fosl2* gene, the super-enhancer in the cKO sample is underscored by a blue line. **e** Most enriched transcription factor binding motifs found in super-enhancers formed in *Med1* cKO compared to Ctrl in CLT tissues; statistical significances are shown as -Log10(*p*-values). All the ChIP-seq data are averages of duplicates conducted in 2 different litters of *Med1* cKO and littermate controls, in which CL tissues are pooled from 2–4 mice for each group (total 4–8 cervical loop tissues) in each ChIP experiment.

Intriguingly, these factors controlling hair lineage were shown to be regulated by super-enhancers in the skin before[4]. Besides the generation of new super-enhancers, we also found an increase in H3K27ac levels around promoters of these genes (Supplementary Fig. 8f), which further positively corelated with an increased gene expression (Fig. 7e). In *Med1* cKO CLH tissues, increased H3K27ac promoter levels positively correlated with an elevated expression of genes involved in preventing ossification (Supplementary Fig. 9b upper arrow and Supplementary Fig. 10a, b), and H3K27ac occupancy increased in ossification inhibiting transcription factors of *Sox9* and *Tob1* loci in their enhancers (Supplementary Fig. 10c). Although we found positive correlations between H3K27ac levels and gene expression, monomethylation of lysine 4 on histone 3 (H3K4me1) may be required to promote actual gene expression[51].

These results further corroborates the dental phenotype of our mouse model which is not only characterized by hair growth but also by enamel dysplasia, and demonstrate that cell fate switch is at least in part due to an inflation of developmentally conserved epidermal and hair enhancers.

In summary, we propose an epigenetic model in which *Med1* safeguards enamel lineage commitment and progression of dental stem cells and their progenies. *Med1* deletion shifts dental stem cells to epidermal fates by amplifying the ectodermal conserved enhancers around hair and epidermal genes.

## Discussion
Adult stem cell fate is heavily linked to their tissue of origin[2,3,52]. It is postulated that the local cellular and endocrine microenvironment primes and directs tissue-specific lineage commitment

and differentiation via cellular, transcription, and epigenetic mechanisms[2,4]. Here we show that deletion of *Med1*, a key component of the enhancer-associated Mediator complex, causes ectopic hair growth on incisors throughout the lifespan of the knockout mice, consistent with continuous incisor growth. Our results suggest that hair growth is driven by a cell fate switch of adult stem cells residing in the incisors. Hair is first generated 4 weeks postnatally which coincides with the point in time when dental stem cells start to regenerate enamel epithelia on incisors. As mouse molars lack adult stem cells, hair may not be regenerated in these teeth. Unlike hair growth in the skin which relies on functionally structured follicles, dental hair is generated from primitive, cellular aggregates which lack typical hair follicle structures. Furthermore, as shown by our depilation experiments, dental hair regrows without induction of the hair cycle, a follicle-dependent process required for regeneration of hair in the skin. Thus, our results question the current knowledge that mammalian hair can only be generated from hair follicles consisting of multiple cell layers bearing defined functions. In addition, our study also illustrates how deletion of *Med1* induces a lineage shift in vivo. Despite that the cellular agglomerates which cause hair growth on incisors exhibit a scattered cellular setup, the mRNA expression profiles found in these structures mimic those of dedicated hair and epidermal cells found in skin epithelia. Thus, while *Med1* negative DESCs reside in the dental environment, they are driven towards lineages characteristic for skin epithelia.

Lineage commitment and differentiation are orchestrated by epigenetic programs and regulatory elements including transcription factors and enhancers[4,29,49]. Our results elucidate that

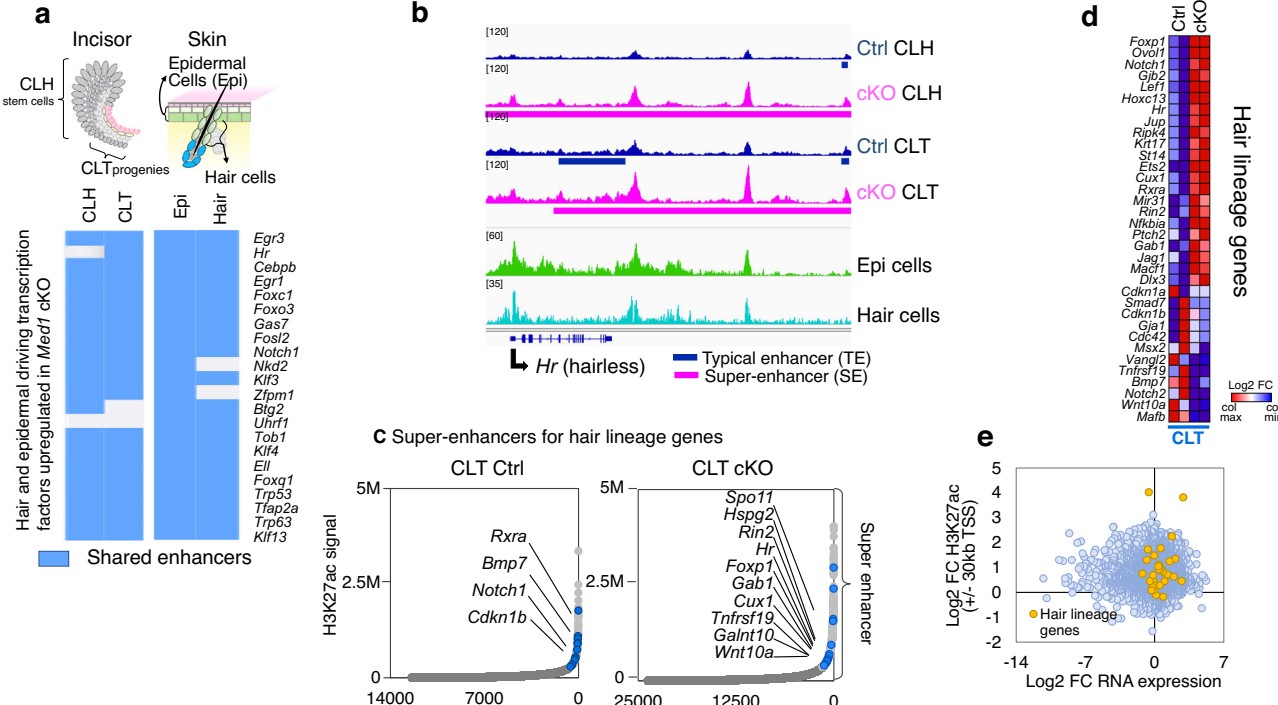

**Fig. 7 Loss of *Med1* advances pre-existing enhancers to super-enhancers near epidermal and hair lineage genes. a** Top, Schematic representations of cell sources for dental and skin epithelia are shown. Bottom, heatmap showing shared enhancers (either TE or SE) between dental (CLH, CLT) and skin (epidermal keratinocytes (Epi), and transient amplifying (TAC) hair follicle keratinocytes (hair)) epithelia for epidermal and hair lineage related transcription factors that are upregulated in *Med1* cKO. **b** ChIP-seq profiles (mm10 genome) around the *Hr* (hairless) locus in CLH and CLT from *Med1* cKO (pink) and Ctrl (blue) mice, compared to epidermal cells (green) and hair TAC keratinocytes (light blue) from skin. Pre-existing enhancers in Ctrl (blue bar) develop into super-enhancers (pink bar) upon loss of *Med1* cKO in CLT. **c** Enhancer distribution profiles; super-enhancers associated with hair lineage genes and exclusively found in Ctrl or *Med1* cKO CLT tissues are noted by the name of neighboring gene (blue circles). **d** Heatmap depicting differential gene expression of hair lineage genes in *Med1* cKO compared to Ctrl in CLT. **e** Correlation between gene expression and H3K27ac promoter occupancy (TSS ± 30 kb) in *Med1* cKO vs Ctrl in CLT tissues for the hair lineage driving gene set (orange dots) compared to all the other genes (blue dots). FC fold change.

already at early dental differentiation stages MED1 containing super-enhancers selectively populate genomic loci coding for enamel fate driven transcription factors in DE-SCs and their progenies. This outlines how *Med1* controls and programs enamel epithelia's differentiation at multiple stages as it primes DE-SCs towards the dental lineage via activation of *Nkx2-3, Pitx2,* and *Isl1* and calibrates lineage progression of DE-SC progenies by regulating the expression of *Runx1/2* and *Satb1*. These findings are consistent with previous studies that *Nkx2-3* controls cusp formation in embryonic dental epithelia[25], and that *Nkx2-3, Pitx2 and Runx2* are essential for dental development[24,25,46]. *Isl1* is required for patterning and mineralization of enamel[26], Runx2 for enamel mineralization[46,47], and *Satb1* for cell polarity and enamel matrix secretion in amelogenesis[48].

Intriguingly, although MED1 is a well-recognized super-enhancer component, loss of *Med1* in CL tissues does not compromise super-enhancer formation. These results are consistent with previous biochemical observations showing that MED1 is dispensable for the formation of the Mediator complex as 9 major Mediator subunits are still found in the complex even after MED1 is removed[38]. Our results demonstrated that *Med1* removal significantly widens the genomic enhancer landscape instead. In the dental setting, this does not increase the potency of DE-SCs but rather drives these cells into epidermal linages. This coincides with a shift in the super-enhancer profiles around loci coding for transcription factors like *Hr* (hairless) and *Rin2* which are essential for hair formation[27,28] as well as *Egr1/3, CEBP/a, CEBP/b,* and *Fosl2* which control epidermal differentiation in the

skin[53,54]. Furthermore, our data show that Tp63/53 binding sites are enriched in super-enhancers in *Med1* cKO mice. These are master regulators for the development of stratified epithelia in vivo[20] and drive epidermal differentiation in vitro. In fact, *Med1* null mice activate Tp63 driven epidermal differentiation in the skin, resulting in an accumulation of epidermal markers and lipids in upper hair follicle keratinocytes resulting in hair loss[33].

Building on our previous studies[39], these results elucidate a central role for Med1 in assuring enamel lineage. Our results also reveal that basal enhancers are already present at hair and epidermis driving transcription factor loci in dental epithelia suggesting that a basal epigenetic memory granting multipotency is already established in pre-occurring ectoderm during embryonic development. Consequently, Med1 guarantees adult stem cells to commit to enamel lineages by restricting super-enhancer formation to lineage-specific loci. In addition, as loss of *Med1* causes DE-SCs to differentiate into a developmentally related lineage, reprogramming of DE-SCs into lineages originating from the same germ layer (ectoderm) might be easier than overcoming more strict and embryonically set epigenetic barriers[29].

Hair and tooth are difficult organs to regenerate through pluripotent stem cells based regenerative approaches, so that no clinical trials have been reported even after extensive efforts[55]. A deep understanding of the cellular and epigenetic mechanisms controlling these lineages and their reprogramming possibilities creates new entry points for the development of strategies to combat hair and tooth-related diseases. Our results establish that hair formation is achievable in tissues outside of the skin, even

without the presence of hair follicles. Similarly, hair was also observed in dental tissues in other mouse models (Sox21 KO[56], Stim1-R304W mutation[42], and Fam83H-KO[41]) and in a few clinical cases showing clinical relevance of this study[43]. Hair was also accidentally generated after DE-SCs were transplanted into renal capsules during an effort to regenerate enamel[57]. This suggests that hair could be generated via genetically engineered cell clusters which might be easier and faster than traditional strategies. This would be of great benefit as it would greatly simplify in vitro hair generation and allow hair growth without hair follicles which are very challenging to regenerate[58]. Similarly, enamel regeneration might also be feasible by using foreskin-derived epidermal stem cells. In fact, cultured epidermal stem cells derived from skin have been shown to differentiate into functional ameloblasts in vitro[59]. Overall, these findings, elucidate possibilities to develop strategies to overcome hair and enamel-associated diseases.

## Methods

**Krt14-driven Med1 cKO mice.** Conditional Med1 knockout (cKO) mice were generated as shown in our previous publication[33]. Floxed (exon 8–10) Med1 mice[60] (C57/BL6 background) were mated with keratin 14 (Krt14) promoter driven Cre recombinase mice (The Jackson Laboratory, C57BL/6 background). All mice were housed in a selected pathogen-free barrier environment with ad libitum access to food and water, 12-hour light and dark cycles, temperature between 20–26 C. Genotyping was performed by PCR[33] and Cre negative littermate mice served as controls (Ctrl). We did not distinguish between male and female mice as in the cKO group, both generate equal amounts of hair in their incisors. All the experiments were approved by the Institutional Animal Care and Ethics Committee at the San Francisco Department of Veterans Affairs Medical Center under the protocol number Bikle 20-019. We have complied with all relevant ethical regulations for mouse handling and treatments.

**Dental epithelial stem cell culture.** CL tissue was separated from 10-week Med1 cKO and Ctrl mice, and epithelial tissues were separated and treated by dispase (5 U/ml Stem Cell Technology) to separate mesenchymal tissues. Epithelial tissues are further dissociated by accutase (Sigma Milipore), and cells were plated under low density to generate colonies and maintained with DMEM media supplemented with recombinant EGF 20 ng/ml (R&D), FGF 25 ng/ml (R&D), and 1X B27 supplement (Gibco). After 8 days of culture, the number and size of colonies are evaluated, and RNA is prepared for microarray.

**BrdU staining.** DE-SC are labeled by BrdU and stained by a BrdU staining kit (Abcam) by following manufacturer's instruction.

**Histological analysis.** Mouse mandibles were dissected from Med1 cKO and littermate control (Ctrl) mice and fixed in 4% paraformaldehyde (PFA) at 4 °C overnight and decalcified in 0.5 M EDTA for 2–3 weeks. Subsequently they were dehydrated and embedded in paraffin wax or OCT compounds using standard protocols. The whole mandible was then serially sectioned at a thickness of 7–10 μm (either sagittal or cross sections). Alternatively, hair generating tissues were dissected from Med1 cKO mice (approximately 6 months), fixed briefly in 4% PFA for 4 h and processed the same way as above without decalcification. Ultimately, slides were stained with hematoxylin and eosin (HE) and imaged with brightfield microscopy.

**Immunofluorescence on paraffin sections.** Paraffin-embedded mandible sections were pretreated in 10 mmol/L citrate buffer (pH 6.0, Sigma-Aldrich) for 20 min using a microwave for antigen retrieval. The specimens were then blocked by Power Block (BioGenex) and incubated with primary antibodies against KRT71 (Progen Lot 208291), LORICRIN (LOR) (Biolegend 905104 former Covance, PRBP-145P-100 D13GF02260 (AF621) Lot 14892401), NOTCH1 (Cell signaling (D1E11) XP® Rabbit mAb Lot 3608 T), and VIMENTIN (Abcam). They were subsequently incubated with species-specific secondary antibodies conjugated to fluorescent dyes (Invitrogen Molecular probes), including Alexa 594 (red) and Alexa 488 (green) and counterstained with DAPI. Images were taken with a confocal microscope (LSM510, Carl Zeiss).

**Immunofluorescence on WT skin and Med1 cKO mandible cryosections.** Frozen sections of whole mandible, or dental hair generating tissues dissected from Med1 cKO mice (6 month), or the dissected back skin from wild-type mice at 4 weeks (anagen) were stained. Skin and tooth tissues were stained by the same protocols in parallel for comparison purposes. These sections were stained with different combinations of two primary antibodies raised against guinea pig or

rabbit to distinguish two antigens with different colors (green and red in secondary reaction) called as double staining. The first primary antibody raised in guinea pig is against hair keratin including KRT71 (same as above 1:100), or KRT75 (ARP, anti-cytokeratin K6HF (K75) Lot 20411-01 1:100), or KRT31 (Progen, Lot 203211 1:100). The second primary antibody raised in rabbit is against either KRT14 (Biolegend 905304 former Covance PRB-155P Lot 14834401 1:1000) or NOTCH1 (same as above 1:200). The sections were subsequently incubated with secondary antibodies containing both Alexa 488 (green) conjugated anti-guinea pig antibody and Alexa 594 (red) bound anti-rabbit antibody (1:1000), then counter stained with DAPI. Different blocking methods using bovine serum or Power Block (BioGenex) were applied. Images were taken with a fluorescent microscope (Zeiss) and processed by ZenPro software.

**RNA expression analysis.** Total RNA was isolated from 4-week-old total CL, CLH, CLT, Sec, or Mat tissues of Med1 cKO and littermate Ctrl mice using the Pico Pure RNA purification kit (ABI). RNA from cultured dental epithelial stem cells was extracted using the RNeasy mini kit (Qiagen). RNAs were analyzed for quality and purity using the Pico Chip kit on an Agilent 2100 Bioanalyzer (Agilent Technologies) to meet the processing criteria of the UCLA Genomic Core facility for the Illumina array platform (Mouse Ref-8 v2.0 Ambion) or RIN score values for RNA-seq. Microarray data generation and primary microarray data analysis was described[7,39]. Microarray data were used for differential pathway analysis with Ingenuity IPA software (Ingenuity) and results were considered significant for p-values less than 0.05. Heat maps were then generated with MeV software using fold changes Log$_2$(cKO/Ctrl). For RNA-Seq analysis, RNA integrity was analyzed with a 2100 Bioanalyzer with an RNA 6000 kit (Agilent Technologies). RNA-Seq libraries were prepared using the Illumina TruSeq v2 library preparation kit and sequenced with paired-end 50 bp reads on an Illumina HiSeq 4000 in biological duplicates.

For quantitative real-time polymerase chain reaction (RT-qPCR), cDNA was synthesized from ~0.5 μg RNA using the SuperScript III first strand synthesis system (Invitrogen) or cDNA Reverse Transcription Kit with RNase Inhibitor (ABI Thermofisher) as described by the supplier and subjected to RT-qPCR amplification using the QuantiTect SYBR-Green PCR Kit (Qiagen) or Powerup SYBR green master mix (ABI Thermofisher) in 7500 or QuantStudio Real Time PCR system (Applied Biosystems) using the following conditions: 10 min 95 °C followed by 45 cycles of 30 s at 95 °C, 30 s at 60 °C and 30 s at 72 °C. Relative mRNA levels were compared to the housekeeping gene Gapdh and determined with the ΔΔCt method. Primers for the analyzed genes are derived from PrimerBank. All RT-qPCR assays were performed in triplicates.

**Chromatin immunoprecipitation.** Chromatin IP was performed by using the LowCell# ChIP kit (Diagenode) for H3K27ac, and the iDeal ChIP-kit (Diagenode) for MED1 according to the manufacturer's protocols with some modifications as described below. Dissected CLH or CLT tissues were cross-linked with 1% paraformaldehyde for 8 min (H3K27ac) or for 15 min (MED1) and subsequently quenched with 0.125 M glycine. Epidermal keratinocytes containing stem and their progenies were isolated from mouse skin were fixed with 1% paraformaldehyde for 10 min. Chromatin was then sonicated with a Covaris S2200 ultra-sonicator (Covaris, Inc.) to obtain DNA fragments with an average length of 300 ± 50 bp. The fragment size was verified with an Agilent 2100 Bioanalyzer (Agilent Technologies). Sheared chromatin was thereafter immunoprecipitated with Protein A-coated magnetic beads (Diagenode) preincubated with 4.5 μg/IP MED1 antibody (former Bethyl laboratories, Thermo Fisher A300-793A lot 9,10), or 3 μg/IP H3K27ac antibody (ab4729, Abcam, lot GR3211959-1 and GR3303561-2). Chromatin input samples were used as reference. Complexes were then washed and eluted from the beads with washing buffer and crosslinks were reversed by incubation at 65 °C for 4 h. Immunoprecipitated DNA along with genomic DNA (Inputs) were purified using the IPure v2 kit (Diagenode). IP efficiency was confirmed by qPCR using primers provided in the kits. For each experiment, CL tissues (right and left) were dissected from 2–4 mice from Med1 cKO and littermate controls and pooled (total 4–8 CL tissues) before immunoprecipitation. We then repeated the Chip-seq experiments using a different litter using the same approaches. ChIP-Seq libraries were generated using the Accel-NGS 2S Plus DNA library kits (Swift Sciences) and amplified by PCR for 11–13 cycles. High molecular weight smear was removed in the library by right side size selection using SPRI beads (Beckman Coulter). The libraries were quantified by Agilent 2100 Bioanalyzer with the high sensitivity DNA assay and sequenced on an Illumina HiSeq 4000 (UCSF Center of advanced technology) in a single strand 50 bp run.

**Next generation DNA sequence data analysis.** For RNA-Seq data analysis, raw reads were assessed for run quality using fastqc version 0.72 followed by alignment to reference genome (mm10) using RNA STAR Galaxy version 2.6.0b-1 with default settings. Differential gene expression analysis was performed using feature Counts Galaxy version 1.6.3 and DESeq2 Galaxy version 2.11.40.2. For ChIP-Seq data analysis, raw reads from MED1 or H3K27ac ChIP-seq sequencing experiments were assessed for quality using fastqc version 0.72, followed by mapping to reference genome (mm10) using BWA Galaxy version 0.7.15.1 with standard settings. Pooled peak calling was performed with MACS2 callpeak version 2.1.0.20140616.0 using broad peak settings and a minimum FDR cutoff of 0.05.

Raw-files for available H3K27ac ChIP-Seq datasets in public from wild-type mice hair stem cell progenies were downloaded from the NCBI-SRA database (SRR1573273 and SRR1573274) and analyzed as described above. To assess significantly different peak occupancy between two experimental groups, peak files (MACS2) were analyzed with DiffBind Galaxy version 2.6.6.4 or 2.10.0. Peak size and location were visualized with IGV version 2.3.92. MAYO using group auto-scale for comparable visualization of all shown groups. Peak intensity around TSS (±10 kbp) as well as genomic peak distribution was assessed with CEAS software (Version1.0.0, Cistrome, Liu Lab). Gene-ontology analysis for MED1 or H3K27ac distal intergenic cis-regulatory elements was performed using GREAT version 3.0.0 (Bejerano Lab, Stanford). Super enhancer analysis was performed using the ROSE package and the software NaviSE. Overlapping typical and super-enhancers between the shown comparisons were assessed with Bedtools (version 2.29) using a minimum overlap of 50%. Motif analysis (de novo and known) was performed using HOMER software. Curated gene-sets used for analyses in this study are provided upon requests.

**Statistics and reproducibility.** All the experiments using *Med1* cKO and littermate control mice ($n = 3$) were repeated with at least two litters, and reproducibility was confirmed. The experiments using cultured dental epithelia were conducted in duplicates for microarray, and reproducibility was confirmed. Histologic analyses were performed using 2 sections per mouse per each group of cKO and Ctrl, and representative images are shown. Statistical significance was calculated using software integrated methods or two-tailed unpaired Student's *t* test. If not differently noted, differences with a *p*-value of <0.05 were considered statistically significant.

**Reporting summary**. Further information on research design is available in the Nature Portfolio Reporting Summary linked to this article.

## Data availability

The array data were previously submitted to a public database (GEO/NCBI/NIH http://www.nlm.nih.gov/geo) and are available with accession numbers GSE50503 (GSM1220311 to GSM1220316) which are part of the under super-series GSE50504 at URL: https://www.ncbi.nlm.nih.gov/geo/query/acc.cgi?acc=GSE50501. The ChIP-seq data are available with accession number GSE221565. RNA-seq data are available under the GEO accession number GSE232190 at the following URL: https://www.ncbi.nlm.nih.gov/geo/query/acc.cgi?acc=GSE232190. Source data underlying figures are presented in Supplementary Data 1.

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

## Acknowledgements

We thank Ms. Sun Hee Kim Wong for maintaining the mice and technical assistance. We are also grateful to Dr. L. Hu, C. Fong for technical support. We also thank Dr. A.J. Van Wijnen for support in epigenetic questions. This work was supported by the NIH grant R21 DE025357 (Y.O.), R01 AR050023 (D.D.B.), DOD grant CA110338 (D.D.B.), and VA Merit I01 BX003814-01 (D.D.B.).

## Author contributions

Conceptualization: Y.O., R.T., K.Y., S.F., P.D., and D.D.B; Methodology: Y.O., R.T., T.N., and K.Y.; Investigation: Y.O., R.T., T.N., and K.Y.; Writing-original draft: Y.O. Writing-review and editing: R.T., Y.O., S.F., P.D., and D.D.B; Funding acquisition: D.D.B., Y.O., S.F., and K.Y., Resources: R.T., K.Y., S.F., D.D.B., and Y.O.

## Competing interests

The authors declare no competing interests.
