## [Peer Review File · Communications Biology]

Reviewers' comments:

Reviewer #1 (Remarks to the Author):

Comments to the Author

The authors investigate the underlying mechanism of enamel-hair lineage transition induced by Mediator 1 ablation. This study has very interesting and informative potential. Using approaches RNA-seq and CHIP-seq analysis, the authors demonstrate that this intriguing phenomenon is due to a reshape of the enhancer landscape in which ectoderm conserved enhancers are amplified to induce epidermal and hair driving transcription factors.

Although bioinformatics analysis is a good prediction method, but some conclusions should be further confirmed by experiments.

Major comments

1. Fig.5 d and f

The authors found by RNA-seq analysis that ablation of MED1 triggered down-regulation of key transcription factors (Satb1, Runx1 and Runx2 etc.). Nevertheless, a set of transcription factors involved in epidermal differentiation of the skin were substantially upregulated in Med1 cKO mice at CLT tissues.

In order to confirm the results more accurately, the RNA-seq analysis results should be confirmed by QPCR.

2. L129-132

Deletion of Med1 from Krt14 expressing DE-SCs and their progenies provoked a major phenotypical shift in the dental compartment causing enamel formation to be replaced by unusual, ectopic hair growth on mouse incisors.

Does the same thing happen in molars? If not, have you ever thought about the reasons behind it.

3. L290-292

Dental epithelia widely share the enhancers with skin epithelia both epidermis and hair follicle derived cells for most of the transcription factors induced upon Med1 loss.

Is the conclusion based on the author's experimental results or the literature?

Here, does the shared enhancer refer to the type, the number or the size of enhancers ?

4. CHIP-seq analysis of H3K4me1 in Fig.6d and Fig.7b

Most H3K4me1 modifications are enriched in enhancer regions. H3K27ac is associated with gene activation and is mainly enriched in enhancer and promoter regions, which are in equilibrium when only H3K4me1 modifications are enriched in enhancer regions; When H3K4me1 and H3K27ac are enriched in the enhancer region, the enhancer is activated to promote gene expression.

For the sake of more accurate results, it is recommended to do CHIP-seq analysis of H3K4me1 and H3K27ac, and then analysis

5. Standard deviation of CHIP-seq analysis in Fig.6c,d and Fig.7b

Mouse tissue was used for CHIP-seq analysis, and individual differences should be present. However, Why standard deviation analysis was not present in the experimental results.

Reviewer #2 (Remarks to the Author):

In the manuscript entitled "Mediator 1 ablation induces enamel-to-hair lineage conversion through enhancer dynamics", Thaler et al. investigate the mechanism through which hair grow in the incisors of mice in which the Med1 gene has been deleted in dental epithelial cells. They show that the ectopic

hair growth happens without forming characteristic hair follicles, even though the structure of the hair shafts is normal and anagen can be induced by depilation. Using RNA-seq analysis, the authors demonstrate that the enamel organ of Med1-cKO mice expresses genes that are specific to epidermal differentiation and hair development, a gene expression signature that is also partially observed (epidermal markers) in cells that were isolated from the cervical loop and grown in vitro. Using ChIP-seq analysis, the authors show that Med1 deletion results in a shift of the association of super-enhancers from enamel fate related transcription factors to epidermal differentiation related transcription factors.

The manuscript is well written and presents interesting findings related to the epigenetic control of ectodermal appendage specification. The phenotype itself has already been reported in a previous publication that also included transcriptomic analysis (microarray). The data from the cell culture and the ChIP-seq analysis are novel.

However, there are a few points that need to be addressed:

- 1) The authors claim that hair growth in the enamel organ is not associated with the presence of hair follicles. It is true that the morphology of the base of the hair is very different from what is seen in the skin. However, the different layers that are present in the hair follicle in the skin are probably also present in these "atypical cell clusters" seen in the enamel organ of Med1-cKO mice. The marker K71 (protein produced by the Krt71 gene) is expressed in the inner root sheath (IRS), and K75, which is also overexpressed, is a marker of the companion layer (CL). Outer root sheath, CL and IRS are the parts of the hair follicle bulb that do not contribute to the formation of the hair shaft. They are supportive layers that are obviously present in the enamel organ of Med1-cKO mice. Cross section of the tissue in the area surrounding of the hair roots (as shown in fig 1b) would help visualize these different layers and better characterize the "pseudo hair follicles" that are forming in this model.
- 2) Is the skin, as stated in the introduction, it is well established that hair growth is dependent on the dermal papilla (ectodermal-mesenchymal interaction). The authors do not mention the dermal papilla in their description of the hair growing in the incisor. In their previous publication, the authors reported that the overall level of ALPL in the enamel organ was reduced. As ALPL is a marker of the dermal papilla, I wonder if there could be clusters of ALPL+ cells associated to the base of the hair root. The authors should also check if other markers of the dermal papilla can be detected (CRABP1, Corin, Sox2).
- 3) The hair shafts produced are obviously pigmented which implies that melanocytes, which are located at the periphery of the dermal papilla in the hair follicle, must also be present in the tissue. Marker of melanocytes should be tested. The epigenetic changes caused by the absence of Med1 may be associated with other aberrant lineage specifications/migrations of neural crests that would result in the ectopic presence of melanocytes in the incisor.
- 4) It appears that hair only grows from mandibular incisors and not from molars and maxillary incisors (?). The authors should comment on that and discuss the possible reason why this is.
- 5) There are at least two other animal models in which this phenotype has been observed (Stim1-R304W and Fam83H-KO). These should be discussed.
- 6) As for the epigenetic mechanism, what happens to the Mediator complex, which is a large complex involving multiple proteins, when Med1 is absent? Have the authors performed ChIP-seq analysis using antibodies against other components of the complex, in WT and Med1-cKO mice?
- 7) There are very rare clinical cases of hair growing in the gingiva. One of the most recent one is PMID 31473154. It may be worth mentioning in the discussion.

Reviewer #3 (Remarks to the Author):

The present ms. by Oda and colleagues reports enamel-to-hair lineage conversion as a result of changes enhancer dynamics following Mediator 1 ablation. Mediator 1 is a Mediator complex subunit

involved in linking enhancers to RNAPol at TSSs of eukaryotic gene promoters. The authors report a phenotype in which hairs are generated in dental incisors instead of tooth enamel. This is a highly significant phenotype. Shedding light on Mediator function is a milestone as well. Overall, the manuscript is well written. The impressive phenotype makes the manuscript suitable for consideration in Communications Biology.

Major concerns

1. The authors should provide better documentation of the phenotype, including better histology and improve the sharpness of their images. Higher magnification images may also be preferable. Currently, the phenotype is poorly characterized.

2. There is a lack of a concrete mechanism explaining the phenotype, but this is in line with manuscripts submitted to and published in Communications Biology.

Response to reviewers Manuscript # COMMSBIO-22-3805-T

We sincerely thank the reviewers for expressing their interest in our study and for their time and efforts to review our manuscript. Their insightful and constructive comments, suggestions and thoughts were indeed very valuable to us and helped us to strengthen and refine our previously proposed cellular and epigenetic mechanisms through which loss of Med1 causes hair growth on mouse incisors.

During the last months, we comprehensively and diligently addressed the reviewers' comments and added extensive new *in vivo* data which all support and strengthen our original work. In brief, we significantly expanded our characterization of the incisor phenotype in Med1 cKO mice and now demonstrate that hair on *Med1* cKO incisors feature all the elements found in hair follicle in the skin, but that they are randomly organized in unusual cell clusters (new Suppl. Figures 3 to 6).

These and additional experimentations resulted in 5 new and 2 revised figures, which expanded our manuscript to 7 main figures and 11 supplementary figures. We also carefully reworked our introduction, result and discussion sections to set our new results in context and to address the reviewers' comments and input.

In summary, the newly presented experimental data reinforce our initial work and strengthen our conclusions that Med1 deficiency modulates the enhancer properties causing a switch from the dental epithelial transcriptional program towards hair and epidermis on incisors *in vivo*.

Below are point-by-point responses to the reviewers' comments. The reviewer points are in *italics* and our responses are in plain text. Inserted or revised sentences are shown by underline.

Reviewer #1 (Remarks to the Author):

Comments to the Author

The authors investigate the underlying mechanism of enamel-hair lineage transition induced by Mediator 1 ablation. This study has very interesting and informative potential. Using approaches RNA-seq and CHIP-seq analysis, the authors demonstrate that this intriguing phenomenon is due to a reshape of the enhancer landscape in which ectoderm conserved enhancers are amplified to induce epidermal and hair driving transcription factors.

Although bioinformatics analysis is a good prediction method, but some conclusions should be further confirmed by experiments.

Major comments

Point 1. Fig.5 d and f

The authors found by RNA-seq analysis that ablation of MED1 triggered down-regulation of key transcription factors (Satb1, Runx1 and Runx2 etc.). Nevertheless, a set of transcription factors involved in epidermal differentiation of the skin were substantially upregulated in Med1 cKO mice at CLT tissues. In order to confirm the results more accurately, the RNA-seq analysis results should be confirmed by QPCR.

Response: Thanks for the comment. For key RNA-Seq data, including the expression of primary dental and hair lineage factors in control and Med1 cKO incisors, we now confirmed the results by RT-qPCR. These results are shown in a new panel in Suppl. Fig. 7d.

Point 2. L129-132

Deletion of Med1 from Krt14 expressing DE-SCs and their progenies provoked a major phenotypical shift in the dental compartment causing enamel formation to be replaced by unusual, ectopic hair growth on mouse incisors.

Does the same thing happen in molars? If not, have you ever thought about the reasons behind it.

Response: This is an interesting and valid point which was also brought up by Reviewer #2. As noted above, in our model hair is indeed only formed on incisors but not molars. Interestingly, in mice only incisors contain specific adult stem cells that regenerate the enamel organ throughout their lifespan. Thus, our interpretation is that upon loss of Med1, hair is only formed on incisors due to the cell fate switch of the dental stem cells which are only found on the incisor. We now include and discuss these points (new sentences are underlined).

(Introduction Line number 65-66, Discussion 374-379)

Here we show that deletion of *Med1*, a key component of the enhancer associated Mediator complex, causes ectopic hair growth on incisors throughout the lifespan of the knockout mice, consistent with continuous incisor growth. Our results suggest that hair growth is driven by a cell fate switch of adult stem cells residing in the incisors. Hair is first generated 4 weeks postnatally which coincides with the point in time when dental stem cells start to regenerate enamel epithelia on incisors. As mouse molars lack adult stem cells, hair may not be regenerated in these teeth.

Point 3. L290-292

Dental epithelia widely share the enhancers with skin epithelia both epidermis and hair follicle derived cells for most of the transcription factors induced upon Med1 loss.

Is the conclusion based on the author's experimental results or the literature?

Here, does the shared enhancer refer to the type, the number or the size of enhancers?

Response: We apologize for the misunderstanding. This conclusion is based on our results presented in Fig. 7a and refers to a comparison between H3K27ac based enhancers between

the mentioned tissues. For this purpose, in addition the Chip-seq datasets that we generated, we obtained Chip-seq data sets for the hair follicle cells from a publicly available database and analyzed the samples accordingly to generate the shown comparison. To make this point clearer, we now added a more accurate description of our analysis and results in the corresponding sections in our manuscript (underlined sentences). In Fig 7a, we show the presence of either typical enhancer (TE) or super-enhancers (SE) that are associated with the listed gene-sets (e.g. DESC gene-set, hair lineage gene-set). These enhancers are identified by the location and size of the TE/SEs (We assess the number of SEs per corresponding gene in Suppl. Fig. 8 e).

(Results Line number 337-338)

Through analyses of the H3K27ac enhancer profiles of epidermal keratinocytes and hair follicle cells from published data sets¹, we further found that even though dental epithelia usually, we further found that even though(Fig. 7a).

Point 4. *CHIP-seq analysis of H3K4me1 in Fig.6d and Fig.7b*

Most H3K4me1 modifications are enriched in enhancer regions. H3K27ac is associated with gene activation and is mainly enriched in enhancer and promoter regions, which are in equilibrium when only H3K4me1 modifications are enriched in enhancer regions; When H3K4me1 and H3K27ac are enriched in the enhancer region, the enhancer is activated to promote gene expression.

For the sake of more accurate results, it is recommended to do CHIP-seq analysis of H3K4me1 and H3K27ac, and then analysis

Response: We thank the reviewer for the insights on H3K4me1. Although we acknowledge the validity of this point, after thoughtful editor's recommendation. we now discuss this limitation in the discussion section (underlined) by referring the publication for H3K4me1.

(Results Line number 356-357)

Although we found positive co-relations between H3K27ac levels and gene expression, H3K4me1 modification may be required to promote actual gene expression².

Point 5. *Standard deviation of CHIP-seq analysis in Fig.6c,d and Fig.7b*

Mouse tissue was used for CHIP-seq analysis, and individual differences should be present. However, Why standard deviation analysis was not present in the experimental results.

Response: Thanks for the comment. We prepared these figures and the samples according to the standard of the literature. In our experiments, we pooled dental tissues from 2-4 mice for *Med1* cKO littermate controls (right and left cervical loops, total 4-8 tissues per group) for one immunoprecipitation to avoid individual differences. To avoid batch effects, we repeated the Chip-seq experiments using an independent litter using the same approach. Then, the quality of each

litter-replicate's IPs was assessed by Chip-qPCR and after sequencing we completed diverse quality control steps (e.g. Fastqc version 0.72). We further repeated the experiment if duplicate is not consistent and not pass to our quality control. We now better explain these details in the corresponding materials and methods section and figure legend (Fig. 6). Due to the nature of how genomic Chip-seq profiles are generally presented, it is not possible to show standard deviations (Fig. 6d, Fig 7b) for each peak on promoters or super-enhancers as shown in all the other epigenetic publications. The number of typical enhancer (TE) or super-enhancers (SE) was also calculated through averaged Chip-seq peaks (duplicates) using NaviSE software in which statistical analysis was performed by integrated statistical method but not SD analysis. Therefore, our data does not include individual differences. We now show these numbers as a Table but not bar graph (Fig. 6c).

Method line number 490-492

For each experiment, CL tissues (right and left) were dissected from 2-4 mice from *Med1* cKO and littermate controls and pooled (total 4-8 CL tissues) before immunoprecipitation.

Figure legend Fig. 6 (line number 822-825)

All the Chip-seq data are averages of duplicates conducted in 2 different litters of *Med1* cKO and littermate controls, in which CL tissues are pooled from 2-4 mice for each group (total 4-8 cervical loop tissues) in each Chip experiment.

Reviewer #2 (Remarks to the Author):

In the manuscript entitled “Mediator 1 ablation induces enamel-to-hair lineage conversion through enhancer dynamics”, Thaler et al. investigate the mechanism through which hair grow in the incisors of mice in which the Med1 gene has been deleted in dental epithelial cells. They show that the ectopic hair growth happens without forming characteristic hair follicles, even though the structure of the hair shafts is normal and anagen can be induced by depilation. Using RNA-seq analysis, the authors demonstrate that the enamel organ of Med1-cKO mice expresses genes that are specific to epidermal differentiation and hair development, a gene expression signature that is also partially observed (epidermal markers) in cells that were isolated from the cervical loop and grown in vitro. Using ChIP-seq analysis, the authors show that Med1 deletion results in a shift of the association of super-enhancers from enamel fate related transcription factors to epidermal differentiation related transcription factors.

The manuscript is well written and presents interesting findings related to the epigenetic control of ectodermal appendage specification. The phenotype itself has already been reported in a previous publication that also included transcriptomic analysis (microarray). The data from the cell culture and the ChIP-seq analysis are novel.

However, there are a few points that need to be addressed:

Point 1 *The authors claim that hair growth in the enamel organ is not associated with the presence of hair follicles. It is true that the morphology of the base of the hair is very different from what is seen in the skin. However, the different layers that are present in the hair follicle in the skin are probably also present in these “atypical cell clusters” seen in the enamel organ of Med1-cKO mice. The marker K71 (protein produced by the Krt71 gene) is expressed in the inner root sheath (IRS), and K75, which is also overexpressed, is a marker of the companion layer (CL). Outer root sheath, CL and IRS are the parts of the hair follicle bulb that do not contribute to the formation of the hair shaft. They are supportive layers that are obviously present in the enamel organ of Med1-cKO mice. Cross section of the tissue in the area surrounding of the hair roots (as shown in fig 1b) would help visualize these different layers and better characterize the “pseudo hair follicles” that are forming in this model.*

Response: We thank the reviewer for this detailed and insightful comment. To address this important point, we localized and compared the expression of hair keratins KRT71 (IRS), KRT75 (companion layer, Cl) and KRT31 (hair cortex) in skin hair follicles of control mice versus incisor tissue of *Med1* cKO mice. Indeed, our new results clearly confirm the reviewer’s hypothesis about the presence of different hair keratins (representing distinct hair follicle layers in skin) also in the atypical cell clusters found on the incisors in our mouse model. However, they appear in an unorganized and random fashion differing from skin hair follicles. These new results are presented in the new Suppl. Fig 3 and 4 and to describe these new findings in the result section (underlined).

(Results Line number 170-190)

To better understand dental hair growth and regeneration, we compared hair follicles of the skin, including distinct hair follicle layers, mesenchymal cells such as dermal papilla, and the presence

of melanocytes to pigment the hair, with the cell clusters found around dental hair in *Med1* cKO mice.

In the skin, hair follicles have distinct cellular layers. Using 4 different antibodies, these layers were well distinguishable in cross sections of normal skin hair follicles (4wk, anagen). They form ring structures, in which the ORS layer (KRT14, red) was outmost, followed by a KRT75 positive CL layer and an internal KRT71 expressing IRS layer. The innermost layer was positive for KRT31 (Suppl. Fig. 3b, left 3 panels). The arrangement and spatial distancing from KRT14 positive cells (ORS) were also observable in sagittal sections (Suppl. Fig. 3b far right panels). (legend is available in supplemental information)

Suppl Fig. 3

Next, we assessed if these organized structures can also be found around the basis of dental hair in *Med1* cKO mice. Here, cells expressing KRT75, KRT71 or KRT31 were found as well but they were scattered and did not form follicle structures (Suppl. Fig. 4), thus clearly differing from the skin. Also, no spatial distancing from KRT14 positive cells was observable in sagittal sections (Suppl. Fig. 4b, yellow triangles upper panels) nor ring-like structures were found at cross sections (Suppl. Fig. 4b, lower panels).

The figure legend is sited in supplemental information and higher resolution images are available by downloading them through google drive below, together all the other supplemental figures.

https://docs.google.com/presentation/d/1Owf9Q1cvACkce_t1lukV5OBKzsZG-IZu/edit?usp=sharing&oid=108749921106050571750&rtpof=true&sd=true

Suppl Fig. 4

These results were further confirmed when cryosections of hair generating tissue dissected from *Med1* cKO mandibles (Suppl. Fig. 5a) were analyzed. Again, three hair keratins (KRT75, KRT71, KRT31) were distributed randomly near hair shafts, but did not organize into hair follicle structures (Suppl. Fig. 5b). (details are available in google drive sharing)

Suppl Fig. 5

We also mentioned that different hair keratins were induced in *Med1* cKO through transcriptome analysis (Fig. 3b right) as the reviewer pointed out. We added following sentences.

(Results Line number 229-231)

Different hair keratin genes to represent distinct layers of hair follicles were upregulated, including KRT75 for supportive companion layer, KRT31 for hair cortex, and KRT71 for IRS (Fig. 3b right panel).

Point 2 *Is the skin, as stated in the introduction, it is well established that hair growth is dependent on the dermal papilla (ectodermal-mesenchymal interaction). The authors do not mention the dermal papilla in their description of the hair growing in the incisor. In their previous publication, the authors reported that the overall level of ALPL in the enamel organ was reduced. As ALPL is a maker of the dermal papilla, I wonder if there could be clusters of ALPL+ cells associated to the base of the hair root. The authors should also check if other markers of the dermal papilla can be detected (CRABP1, Corin, Sox2).*

Response: Again, these are very valid and interesting points, and we fully concur with the Reviewer's comments. We added following results (underlined).

(Results Line number 191-198)

In regard to the presence of dermal papilla, such structures were also missing in the roots of incisor-grown hair lacking hair bulbs (HE stains in Fig 1c, Suppl. Fig. 4a). Nevertheless, we found the mesenchymal protein VIMENTIN (red) in KRT71 expressing hair generating cell clusters in *Med1* cKO (6 month) (Suppl. Fig. 5c yellow triangles). Although VIMENTIN does not represent a classic marker for dermal papilla, it was show to localize in this structure in the skin³. However, VIMENTIN expressing cells were scattered and did not form well organized structures, suggesting lack of papilla-like structures in *Med1* cKO mice.

(legend of Fig 5c is available in supplemental information)

During our studies, it has been challenging to identify specific dermal papilla markers in the context of our model as ALPL and SOX2 are strongly expressed in dental epithelial stem cells and SI cells as well. Our study indicates that *Med1* lacking SI epithelia (may express SOX2 or ALPL) is invaded by dental mesenchyme, where hair is generated. Therefore, the expression of SOX2 and ALPL may not allow to distinguish potential dermal papilla cells from dental epithelia. We confirmed that SOX2 is not well expressed in the hair generating region on *Med1* cKO incisors (see images below, a, b). We also found that several dermal papilla markers (e.g. (*Alpl*, *Itga8*, *Sox18*, *Sox2*, *Dcc*, *Vmac (vimentin)*, *Bmp4*, *Crabp1*) are expressed in dental tissues but were not induced upon loss of *Med1* in CLT tissues and rather decreased, except *Corin* (RNA-seq data, see image below). Potential ectodermal-mesenchymal interactions remain to be elucidated in the

future. Thus, as these data are too inconclusive to support the presence of specific dermal papilla markers in our *Med1* cKO context, we did not incorporate the results shown below in the new version of the manuscript and only present them here for clarification.

a, Schematic presentation to show the location of cervical loop (CL), secretory region (Sec) and hair generating region of *Med1* cKO incisor. **b**, SOX2 immunostaining in these 3 regions (SOX2 brown with blue counterstaining). Strong nuclear signals for SOX2 in CL, dental epithelia SI cells and oral epithelia are shown by brown staining, although no staining was observed in hair generating region of *Med1* cKO. Graph, mRNA expression of typical dermal papilla markers (RNA-seq). Percentage expression of cKO CLT compared to littermate control (blue) are shown.

Point 3 The hair shafts produced are obviously pigmented which implies that melanocytes, which are located at the periphery of the dermal papilla in the hair follicle, must also be present in the tissue. Marker of melanocytes should be tested. The epigenetic changes caused by the absence of *Med1* may be associated with other aberrant lineage specifications/migrations of neural crests that would result in the ectopic presence of melanocytes in the incisor.

Response: We thank the reviewer for bringing up the question about the presence of melanocytes. This is a topic which clearly belongs in and reinforces the scientific value of our manuscript. To answer this question, we assessed for melanin accumulation in the pocket tissues present on *Med1* cKO incisors. We found that brown colored, melanin producing cells were again randomly distributed (see new Suppl. Fig. 6a, yellow arrows). This pattern differs from the skin, where melanin is discretely accumulated inside of the hair bulb (new Suppl. Fig. 6b, 4wk anagen). In addition, the mRNA expression of melanocyte markers was elevated in *Med1* cKO CLT tissue

(Suppl. Fig. 6c, RNA-seq) supporting the presence of melanocytes. In the context of these new results we also discuss the possibility that the dental melanocytes in our model may originate from an aberrant lineage conversion of neural crest cells that are capable of regenerating different type of mandibular cells including dental mesenchyme⁴.

We added following sentences (Results Line number 199-208)

Next, we assessed melanin accumulation in the pocket tissues of *Med1* cKO incisors to generate well pigmented black hair (C57BL6 background) (Suppl. Fig. 6a, yellow arrows). Brown colored melanin producing cells were randomly distributed (Suppl. Fig. 6a, enlarged images) again differing from the skin in which the melanin is discretely accumulated inside of the hair bulbs (Suppl. Fig. 6b, 4wk anagen). In addition, mRNA expressions of melanocyte markers were elevated in incisors of *Med1* cKO mice (Suppl. Fig. 6c, RNA-seq) indicating the presence of melanocyte-like cells. We can speculate that this type of cells might be generated through an aberrant lineage conversion of neural crest cells, multipotent stem cells that regenerate mandibular cells including dental mesenchymal cells⁴.

(details are available google drive sharing listed above)

Suppl Fig. 6

Point 4 It appears that hair only grows from mandibular incisors and not from molars and maxillary incisors (?). The authors should comment on that and discuss the possible reason why this is.

Response: This is an interesting and valid point which was also brought up by Reviewer #1. Our answer is shown in above section: Reviewer 1, point 2: L129-132.

Point 5 There are at least two other animal models in which this phenotype has been observed (*Stim1-R304W* and *Fam83H-KO*). These should be discussed.

Response: We thank the Reviewer for this valuable information and apologize for having missed this important literature. We now include these models in our revised manuscript. (Results Line number 134-136, Discussion Line 438-440)

This phenomenon has also been observed in two further unrelated mouse models^{5,6}

Similarly, hair was also observed in dental tissues in other mouse models (*Sox21* KO⁷, *Stim1-R304W* mutation⁶, and *Fam83H-KO*⁵)...

Point 6 As for the epigenetic mechanism, what happens to the Mediator complex, which is a large complex involving multiple proteins, when *Med1* is absent? Have the authors performed ChIP-seq analysis using antibodies against other components of the complex, in WT and *Med1-cKO* mice?

Response: This is an interesting question. As previous studies have shown, MED1 is dispensable for the formation of the Mediator complex. In fact, when nuclear extracts are depleted from MED1 protein, the integrity of the Mediator complex is not affected as 9 of the other major components of Mediator were still found to form the complex (see attached IP-blot)⁸ (data inserted). Further, biochemical studies showed that only 20% of the Mediator complexes contain MED1. These insights can be applied to our work as the Mediator is conserved in different cells, tissues and even organisms, from yeast to mammals. We have already referred this publication, but we now better discuss these aspects in the discussion. Chip-seq may not be practically possible as we did not find Chip qualified antibodies for 25 Mediator subunits.

Immuno-depletion of MED1 containing Mediator complex.

Immunoblots show the Mediator composition after MED1 is removed from the Mediator complex in nuclear extracts. The presence of MED1 lacking Mediator complex still includes 9 major Mediator subunits, as shown. Untreated nuclear extract and pre-immune samples are shown as controls.

We added the following sentences in discussion (Discussion line number 406-409).

These results are consistent with previous biochemical observations showing that MED1 is dispensable for the formation of the Mediator complex as 9 major Mediator subunits are still found in the complex even after MED1 is removed⁸.

Point 7 *There are very rare clinical cases of hair growing in the gingiva. One of the most recent one is PMID 31473154. It may be worth mentioning in the discussion.*

Response: This is great advice which might suggest a clinical relevance of our proposed mechanisms. We now include this essential literature in our revised manuscript. We added following sentences (Results Line number 134-136, Discussion Line 438-440).

Similarly, hair was also observed in dental tissues in other mouse models (Sox21 KO⁷, Stim1-R304W mutation⁶, and Fam83H-KO⁵) and in a few clinical cases showing clinical relevance of this study⁹.

Reviewer #3 (Remarks to the Author):

The present ms. by Oda and colleagues reports enamel-to-hair lineage conversion as a result of changes enhancer dynamics following Mediator 1 ablation. Mediator 1 is a Mediator complex subunit involved in linking enhancers to RNAPol at TSSs of eukaryotic gene promoters. The authors report a phenotype in which hairs are generated in dental incisors instead of tooth enamel. This is a highly significant phenotype. Shedding light on Mediator function is a milestone as well. Overall, the manuscript is well written. The impressive phenotype makes the manuscript suitable for consideration in Communications Biology.
Major concerns

Point 1. *The authors should provide better documentation of the phenotype, including better histology and improve the sharpness of their images. Higher magnification images may also be preferable. Currently, the phenotype is poorly characterized.*

Response: We concur with the Reviewer's point about the necessity to better characterize the phenotype, a point which was also recommended by Reviewer #2. Please see our answers to reviewer 2 above in which we added new results in Suppl Fig 3, 4, 5, 6. These data strengthen our previous studies to report basic hair generating phenotype^a and enamel dysplasia^b of *Med1* cKO mice.

a. Yoshizaki, K. *et al.* Ablation of coactivator Med1 switches the cell fate of dental epithelia to that generating hair. *PLoS One* **9**, e99991 (2014). <https://doi.org:10.1371/journal.pone.0099991>

b. Yoshizaki, K. *et al.* Mediator 1 contributes to enamel mineralization as a coactivator for Notch1 signaling and stimulates transcription of the alkaline phosphatase gene. *J Biol Chem* **292**, 13531-13540 (2017). <https://doi.org/10.1074/jbc.M117.780866>

Regarding the image resolution of our histology images, we now provide higher resolution images, as detailed below but the resolution was still comprised during the process to prepare the submission PDF file. Original figures and supplement figures are available by downloading them through google drive.

1 PPT file for figures

<https://docs.google.com/presentation/d/14IS0khLL0Z1Eupe7P4pgQ8utekQRFAS9/edit?usp=sharing&oid=108749921106050571750&rtpof=true&sd=true>

2 PPT file for supplemental figures

https://docs.google.com/presentation/d/1Owf9Q1cvACkce_t1lukV5OBKzsZG-IZu/edit?usp=sharing&oid=108749921106050571750&rtpof=true&sd=true

Fig. 1c, higher resolution for HE images is provided.

Fig. 2a, higher resolution for HE images is provided.

Fig. 2b, higher resolution for immunofluorescent images is provided.

Fig. 2c, higher resolution for skin hair follicles images (new pictures) and dental tissue images is provided.

Suppl. Fig. 2, higher resolution for images of mandible and dental tissues (HE) is provided.

New Suppl. Fig. 3, original resolution immunofluorescent images are provided.

New Suppl. Fig. 4, original resolution immunofluorescent and HE images are provided.

New Suppl. Fig. 5, original resolution immunofluorescent images are provided.

New Suppl. Fig. 6, higher resolution HE images are provided.

Point 2. *There is a lack of a concrete mechanism explaining the phenotype, but this is in line with manuscripts submitted to and published in Communications Biology.*

Response: We thank the reviewer for the comment. In this revised version of our manuscript, we strengthen our results by adding new data and discussion.

To the editor and all the reviewers.

In addition to address reviewers' questions, we also added followings.

- a. Methods to produce new supplemental figures, detailed information of antibodies.
- b. Accession numbers for Chip-seq and RNA-seq
- c. Minor editorial changes that do not change the context.

New references

- 1 Lien, W. H. *et al.* Genome-wide maps of histone modifications unwind in vivo chromatin states of the hair follicle lineage. *Cell Stem Cell* **9**, 219-232 (2011).
<https://doi.org/10.1016/j.stem.2011.07.015>

- 2 Kang, Y., Kim, Y. W., Kang, J. & Kim, A. Histone H3K4me1 and H3K27ac play roles in
nucleosome eviction and eRNA transcription, respectively, at enhancers. *FASEB J* **35**, e21781
(2021). <https://doi.org/10.1096/fj.202100488R>
- 3 Morioka, K., Arai, M. & Ihara, S. Steady and temporary expressions of smooth muscle actin in
hair, vibrissa, arrector pili muscle, and other hair appendages of developing rats. *Acta Histochem
Cytochem* **44**, 141-153 (2011). <https://doi.org/10.1267/ahc.11013>
- 4 Chai, Y. *et al.* Fate of the mammalian cranial neural crest during tooth and mandibular
morphogenesis. *Development* **127**, 1671-1679 (2000). <https://doi.org/10.1242/dev.127.8.1671>
- 5 Wang, S. K. *et al.* Fam83h null mice support a neomorphic mechanism for human ADHCAI. *Mol
Genet Genomic Med* **4**, 46-67 (2016). <https://doi.org/10.1002/mgg3.178>
- 6 Gamage, T. H. *et al.* STIM1 R304W in mice causes subgingival hair growth and an increased
fraction of trabecular bone. *Cell Calcium* **85**, 102110 (2020).
<https://doi.org/10.1016/j.ceca.2019.102110>
- 7 Saito, K. *et al.* Sox21 Regulates Anapc10 Expression and Determines the Fate of Ectodermal
Organ. *iScience* **23**, 101329 (2020). <https://doi.org/10.1016/j.isci.2020.101329>
- 8 Zhang, X. *et al.* MED1/TRAP220 exists predominantly in a TRAP/ Mediator subpopulation
enriched in RNA polymerase II and is required for ER-mediated transcription. *Mol Cell* **19**, 89-100
(2005). <https://doi.org/10.1016/j.molcel.2005.05.015>
- 9 Zhurakivska, K. *et al.* An unusual case of recurrent gingival hirsutism. *Oral Surg Oral Med Oral
Pathol Oral Radiol* **129**, e200-e203 (2020). <https://doi.org/10.1016/j.oooo.2019.08.003>

REVIEWERS' COMMENTS:

Reviewer #2 (Remarks to the Author):

The author has answered reviewer's question, I is basically satisfied and agrees to publish.

Reviewer #3 (Remarks to the Author):

The authors have appropriately addressed the comments and suggestions submitted by the reviewer.

Reviewer #4 (Remarks to the Author):

The present manuscript by Dr. Oda and colleagues is an important study related to the role of mediator 1.

All previous critiques have been successfully addressed by this author.